# Waveform detection by deep learning reveals multi-area spindles that are selectively modulated by memory load

Maryam H Mofrad[1,2], Greydon Gilmore[2,3], Dominik Koller[4], Seyed M Mirsattari[5,6,7,8], Jorge G Burneo[5,9], David A Steven[5,9], Ali R Khan[2,3,7], Ana Suller Marti[2,5], Lyle Muller[1,2]*

[1]Department of Mathematics, Western University, London, Canada; [2]Brain and Mind Institute, Western University, London, Canada; [3]Department of Biomedical Engineering, Western University, London, Canada; [4]Advanced Concepts Team, European Space Agency, Noordwijk, Netherlands; [5]Department of Clinical Neurological Sciences, Schulich School of Medicine and Dentistry, Western University, London, Canada; [6]Department of Medical Imaging, Schulich School of Medicine and Dentistry, Western University, London, Canada; [7]Department of Medical Biophysics, Schulich School of Medicine and Dentistry, Western University, London, Canada; [8]Department of Psychology, Western University, London, Canada; [9]Department of Epidemiology and Biostatistics, Schulich School of Medicine and Dentistry, Western University, London, Canada

*For correspondence:
lmuller2@uwo.ca

Competing interest: The authors declare that no competing interests exist.

**Abstract** Sleep is generally considered to be a state of large-scale synchrony across thalamus and neocortex; however, recent work has challenged this idea by reporting isolated sleep rhythms such as slow oscillations and spindles. What is the spatial scale of sleep rhythms? To answer this question, we adapted deep learning algorithms initially developed for detecting earthquakes and gravitational waves in high-noise settings for analysis of neural recordings in sleep. We then studied sleep spindles in non-human primate electrocorticography (ECoG), human electroencephalogram (EEG), and clinical intracranial electroencephalogram (iEEG) recordings in the human. Within each recording type, we find widespread spindles occur much more frequently than previously reported. We then analyzed the spatiotemporal patterns of these large-scale, multi-area spindles and, in the EEG recordings, how spindle patterns change following a visual memory task. Our results reveal a potential role for widespread, multi-area spindles in consolidation of memories in networks widely distributed across primate cortex.

## Editor's evaluation

This article provides compelling evidence that deep convolutional networks can detect repeating patterns in biological data better than existing methods, in the presence of noise, biological or otherwise. In analyses of data acquired from the brains of primates using various modalities, the authors show that spindles in cortex have a wider spatial distribution that previously thought. Applications of the proposed approach in other settings may lead to novel findings about the distribution of transient oscillatory patterns in the brain.

**eLife digest** The brain processes memories as we sleep, generating rhythms of electrical activity called 'sleep spindles'. Sleep spindles were long thought to be a state where the entire brain was fully synchronized by this rhythm. This was based on EEG recordings, short for electroencephalogram, a technique that uses electrodes on the scalp to measure electrical activity in the outermost layer of the brain, the cortex. But more recent intracranial recordings of people undergoing brain surgery have challenged this idea and suggested that sleep spindles may not be a state of global brain synchronization, but rather localised to specific areas.

Mofrad et al. sought to clarify the extent to which spindles co-occur at multiple sites in the brain, which could shed light on how networks of neurons coordinate memory storage during sleep. To analyse highly variable brain wave recordings, Mofrad et al. adapted deep learning algorithms initially developed for detecting earthquakes and gravitational waves. The resulting algorithm, designed to more sensitively detect spindles amongst other brain activity, was then applied to a range of sleep recordings from humans and macaque monkeys.

The analyses revealed that widespread and complex patterns of spindle rhythms, spanning multiple areas in the cortex of the brain, actually appear much more frequently than previously thought. This finding was consistent across all the recordings analysed, even recordings under the skull, which provide the clearest window into brain circuits. Further analyses found that these multi-area spindles occurred more often in sleep after people had completed tasks that required holding many visual scenes in memory, as opposed to control conditions with fewer visual scenes.

In summary, Mofrad et al. show that neuroscientists had previously not appreciated the complex and dynamic patterns in this sleep rhythm. These patterns in sleep spindles may be able to adapt based on the demands needed for memory storage, and this will be the subject of future work. Moreover, the findings support the idea that sleep spindles help coordinate the consolidation of memories in brain circuits that stretch across the cortex. Understanding this mechanism may provide insights into how memory falters in aging and sleep-related diseases, such as Alzheimer's disease. Lastly, the algorithm developed by Mofrad et al. stands to be a useful tool for analysing other rhythmic waveforms in noisy recordings.

## Introduction

Consolidation of long-term memories requires precise coordination of pre- and postsynaptic spikes across neocortex. New memories are transferred from hippocampus to neocortex for long-term storage (*McClelland et al., 1995*; *Rasch and Born, 2007*), where interconnections within a sparse, distributed neuron group are strengthened until their activity becomes hippocampus-independent (*Frankland and Bontempi, 2005*). Computational studies have identified neural oscillations as a potential mechanism to regulate synaptic plasticity (*Masquelier et al., 2009*; *Song et al., 2000*) and create precise spike timing (*Cassenaer and Laurent, 2007*; *Muller et al., 2011*). Further, experiments have shown that the sleep 'spindle' oscillation influences spiking activity (*Contreras and Steriade, 1995*; *Kandel and Buzsáki, 1997*; *Peyrache et al., 2011*) and causally contributes to sleep-dependent consolidation of long-term memory (*Mednick et al., 2013*). It remains unclear, however, precisely how this rhythm can coordinate activity across areas in neocortex for synaptic plasticity and long-term storage to occur.

While early recordings in anesthetized animals (*Andersen et al., 1967*; *Contreras et al., 1996*) and human electroencephalogram (EEG) (*Achermann and Borbély, 1998*) indicated that sleep spindles generally occur across a wide area in cortex, creating a state of large-scale synchrony (*Sejnowski and Destexhe, 2000*; *Steriade, 2003*), recent work in intracranial recordings from human clinical patients has challenged this idea by reporting isolated, 'local' sleep spindles (*Andrillon et al., 2011*; *Nir et al., 2011*; *Piantoni et al., 2017*; *Sarasso et al., 2014*, but see *Frauscher et al., 2015*). Because spindles are intrinsically related to sleep-dependent consolidation of long-term memory (*Clemens et al., 2005*; *Gais et al., 2002*; *Mednick et al., 2013*), this difference in reported spatial extent of the spindle raises an important question for the organization of engrams established through sleep-dependent memory consolidation. Recent evidence using *cFos* mapping in animal models suggests these engrams are distributed widely across brain areas (*Kitamura et al., 2017*; *Roy et al., 2019* n.d.), which is consistent

with previous imaging evidence in the human (*Brodt and Gais, 2021*; *Wheeler et al., 2000*). Taking these points together, we reasoned that widespread, multi-area spindles may occur more often than previously reported in primate and human cortex. If this were the case, these widespread spindles could provide the mechanism needed to link populations distributed widely across the cortex for sleep-dependent memory consolidation.

One potential mechanism is provided by previous work on spindles in intracranial electrocorticography (ECoG) recordings in human clinical patients, where these oscillations were found to be organized into a wave rotating across the cortex (see Video 1 in *Muller et al., 2016*). Based on their speed of propagation (2–5 m/s), which matches the axonal conduction speeds of long-range white matter fibers in cortex, it was identified that this rotating wave organization could precisely align spikes across areas separated by long distances in cortex to create the conditions necessary for both synaptic strengthening and weakening to occur. With this previously identified mechanism in mind, we thus hypothesized that widespread, multi-area spindles might be a critical missing link in understanding how networks widely distributed across cortex are modulated during sleep.

Reliably detecting individual spindles in noisy sleep recordings, however, is challenging. Spindle oscillation amplitudes differ across regions in cortex (*Frauscher et al., 2015*). Furthermore, oscillation amplitudes may differ significantly across recording sites simply due to variation in electrode properties (*Kappenman and Luck, 2010*; *Nelson and Pouget, 2010*). For these reasons, we reasoned that bandpass filtering followed by an amplitude threshold (AT), which is a technique common across methods for spindle detection (*Warby et al., 2014*), may only detect the largest-amplitude events, potentially leading to an underestimation of spatial extent. To address this question, we adapted deep learning algorithms initially developed for detecting earthquakes (*Perol et al., 2018*) and gravitational waves (*George and Huerta, 2018*) in high-noise settings to analysis of neural recordings in sleep. These convolutional neural networks (CNNs) are relatively general to the type of noise in each recording, provided there is enough training data and a set of high-quality marked events. Because obtaining many high-quality marked spindle events is itself difficult, however, as sleep recordings are in general manually scored by experts (a process that is both expensive and subjective), we introduce here a careful, two-step computational approach. First, we use a signal-to-noise ratio (SNR) algorithm (*Muller et al., 2016*) to generate a set of high-quality marked spindles for training the CNN. The SNR algorithm, which is closely related to the constant false alarm rate (CFAR) method used in radar (*Richards, 2005*), detects many 'true' spindles while minimizing false detections. This property makes the SNR algorithm an excellent method for generating a high-quality training dataset and, in addition, for providing a second check on results from the CNN model on the subset of spindles detected by this more conservative approach. We then use the trained CNN to detect a comprehensive set of spindles in sleep recordings. To test this approach, we studied sleep spindles in macaque non-human primate (NHP) ECoG, human electroencephalogram (EEG), and, finally, clinical intracranial electroencephalogram (iEEG) recordings, which provide a window into the circuits of the human brain at one of the highest spatial resolutions possible (*Lachaux et al., 2012*; *Mukamel and Fried, 2012*). This two-step approach results in a subject-specific model, adapted to the noise encountered in each recording type and the specific sleep waveforms in each individual, that can more sensitively detect a range of clearly formed large- and small-amplitude spindles in the sleep recordings. Finally, at each point in the analysis, we also return to the subset of spindles detected by the SNR algorithm to validate results obtained from the CNN.

Our approach reveals that the spatial extent of spindles, defined here in terms of co-occurrence across electrode sites within the same 500 ms detection window, is widely distributed over a broad range of cortex. In particular, multi-area spindles are much more frequent than previously estimated by AT approaches, which tend to select only the highest-amplitude spindles and could miss events that transiently fall below threshold. Importantly, while we apply our approach to very different datasets (ECoG, EEG, and iEEG) in this work, the comparisons we make are always between the spatial extent of spindles detected by our CNN approach and AT methods within an individual recording type. These results provide strong evidence that widespread, multi-area spindles may have been underestimated in previous work. This finding, which clearly emerges consistently across all recording types, is not affected by differences in spatial sampling of different electrode types, as we always restrict comparisons within a single type of electrode. In human sleep EEG after low- (L-VM) and high-load visual memory (H-VM) tasks, our method also detects an increase in regional and multi-area

spindles uniquely following an H-VM task. Finally, we note that spindle co-occurrence does not imply zero-lag synchrony across recording sites, with all sites reaching positive (or negative) peaks in potential at the same point in time. Further spatiotemporal analysis of the sleep EEG recording reveals that the multi-area spindles are organized into rotating waves that are also modulated by the memory task. Taken together, these results reveal a sophisticated spatiotemporal organization of sleep spindles in the primate brain, both in co-occurrence and in phase organization, that has previously gone unappreciated. These results thus provide substantial insight into the spatiotemporal organization of sleep spindles in the primate brain, during normal sleep and also following memory tasks.

## Results

Sleep recordings from both human and NHP were obtained from electrodes ranging from traditional scalp EEG to invasive intracranial EEG electrodes (*Figure 1a*). We trained subject-specific CNN models over high-quality training datasets generated by the SNR algorithm. The SNR algorithm robustly detects spindles ranging from high to lower amplitudes (*Figure 1—figure supplement 1*), providing a good training set for the CNN. To verify the quality of spindles detected by our CNN model (*Figure 1c*), we first computed average power spectral densities (PSDs) over spindle and non-spindle windows. The average PSD of detected spindle events shows an increase in the 11–15 Hz spindle frequency range (red lines, *Figure 1b*), while non-spindle events do not show a corresponding increase (black lines, *Figure 1b*). Spindles detected by the CNN are well formed, consistent with standard morphology (*Loomis et al., 1935*; *Newton Harvey et al., 1937*; *Silber et al., 2007*; *Figure 1d*), and in agreement with previously observed durations (average ± SEM: 0.69±0.004 s, NHP ECoG; 0.87±0.006 s, EEG; 0.77±0.009 s, iEEG) (*Fernandez and Lüthi, 2020*; *Takeuchi et al., 2016*; *Warby et al., 2014*). To further validate spindles detected by the CNN, we designed a time-shifted averaging approach for application to recordings with only a 1 Hz highpass filter applied (thus excluding any potential effects from lowpass filtering). To do this, we collected signals from detected spindles, filtered at a 1 Hz highpass, time-aligned the events to the largest positive value within the detected window (corresponding to a positive oscillation peak), and then computed the average across aligned events. With this approach, the average over detected spindles exhibited clear 11–15 Hz oscillatory structure (black line, *Figure 1—figure supplement 2*), while no oscillatory structure is observed when averaging over time-matched randomly selected non-spindle activity (dashed red line, *Figure 1—figure supplement 2*). This result demonstrates that spindles detected by the CNN exhibit the correct structure even in a mostly raw, unprocessed signal with no lowpass filtering applied, while non-spindle activities only exhibit a peak due to the alignment to the central peak in the window, with a decay consistent with the autocorrelation time present in the 1 Hz filtered signal. We then compared the average number of spindles per minute (*Figure 1—figure supplement 3a*) and the distribution of peak Fourier amplitudes in the 9–18 Hz band for spindle events detected by the CNN and AT approach (*Figure 1—figure supplement 3b*). In the intracranial recordings (ECoG and iEEG), AT detects a subset of spindles that are significantly higher amplitude than those detected by the CNN (p<0.02, NHP ECoG recordings; p<1 × 10⁻¹²; iEEG recordings, one-sided Wilcoxon signed-rank test; n.s. in EEG), consistent with the expectation that AT will preferentially select the largest amplitude events. The CNN, however, detects a broader set of spindles and can find well-formed spindles that are both large and small in amplitude (*Figure 1—figure supplement 4*). This improved resolution allows us to study the spatial extent of spindles in an approximately amplitude-invariant manner. Furthermore, to understand more generally the performance of the CNN and AT approaches under different types of noise and in the presence of artifacts, we conducted a detailed simulation study using surrogate data with systematically varying noise characteristics or rate of artifacts (*Supplementary file 1* and *Figure 1—figure supplements 5 and 6*). Finally, we used the pattern of activation of the feature map and gradient map to study the underlying mechanism by which the trained CNN detects sleep spindle oscillations (*Figure 1—figure supplements 7 and 8*) and we evaluated the choice of architecture tailored with respect to the duration of rhythmic activity (*Figure 1—figure supplement 9*).

What is the spatial extent of spindle oscillations across cortex? To answer this question, we studied the distribution of spindle co-occurrence across electrodes in the sleep recordings. We defined three classes of spindles based on co-occurrence across recording sites: local (1–2 sites), regional (3–10 sites), and multi-area (more than 10 sites). We noted that our CNN approach detected many spindles with electrode sites distributed widely across the cortex (*Figure 2a*). By taking into account the

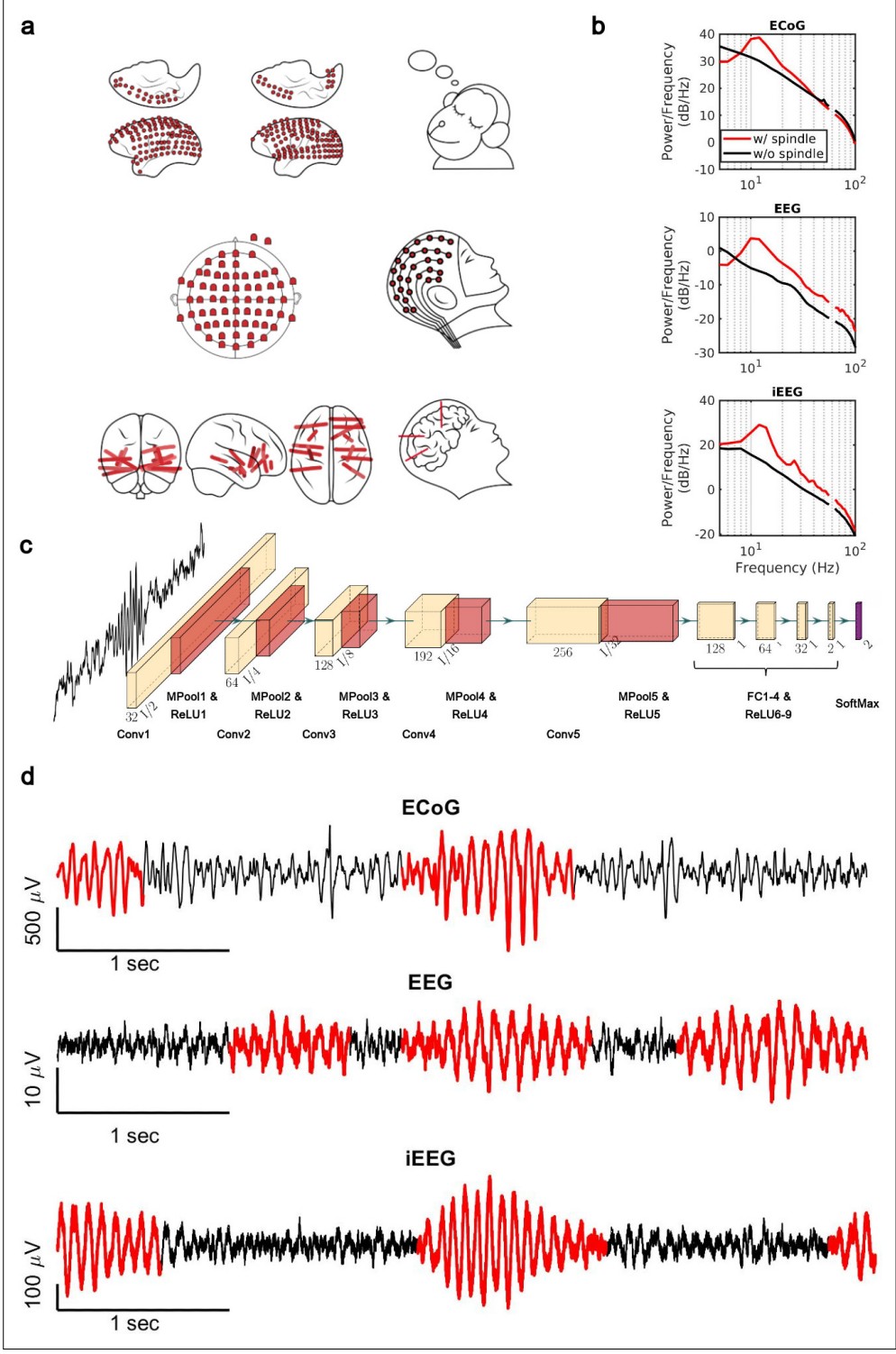

**Figure 1.** Electrophysiology, architecture of the convolutional neural network (CNN) model, and detected spindles. (**a**) Electrode placement of multichannel electrocorticography (ECoG) recordings of two macaques (top), high-density scalp electroencephalogram (EEG) used for recordings after low- and high-load visual memory tasks (middle), and example intracranial electroencephalogram (iEEG) contacts in a human clinical patient (bottom). (**b**) Average power spectral density estimate for spindle windows detected by the CNN model (red) and matched non-spindle windows (black), illustrating the nearly 10 dB increase within the 11–15 Hz spindle band in non-human primate (NHP) ECoG recordings (top), human EEG recordings (middle), and human iEEG recordings (bottom).

*Figure 1 continued on next page*

*Figure 1 continued*

Power at line noise frequency omitted for clarity. (**c**) The architecture of the CNN model developed for spindle detection. (**d**) Examples of detected spindles by the CNN model (red) in NHP ECoG recordings (top), human EEG recordings (middle), and human iEEG recordings (bottom).

The online version of this article includes the following figure supplement(s) for figure 1:

**Figure supplement 1.** Signal-to-noise ratio (SNR) and amplitude-thresholding (AT) sensitivity to the amplitude (surrogate data – simulated spindles).

**Figure supplement 2.** Average time-shifted spindles detected by the convolutional neural network (CNN) model.

**Figure supplement 3.** Average single-electrode spindle frequency and amplitude distribution of convolutional neural network (CNN) model versus amplitude-thresholding (AT) algorithm.

**Figure supplement 4.** Performance of convolutional neural network (CNN) model versus amplitude-thresholding (AT) algorithm.

**Figure supplement 5.** A spindle with varying types of noise detected by the two-step model (surrogate data – simulated spindles).

**Figure supplement 6.** Performance of convolutional neural network (CNN) approach and amplitude-thresholding (AT) with increasing rate of noise artifacts.

**Figure supplement 7.** Feature map.

**Figure supplement 8.** Gradient attribution map.

**Figure supplement 9.** Impact of filter size on the convolutional neural network (CNN) model performance.

**Figure supplement 10.** Power spectral density (PSD) comparison.

**Figure supplement 11.** Impact of signal-to-noise ratio (SNR) threshold on the convolutional neural network (CNN) model.

**Figure supplement 12.** Non-spindle activities detected by the amplitude-thresholding (AT) algorithm.

---

unique cortical regions sampled by electrodes in each individual (9 on average, ranging from 7 to 12 cortical regions, *Supplementary file 2*), we verified that these were indeed multi-area spindle events (*Figure 2b*) that happen on average across 60% of recorded cortical regions. Considering the different spatial sampling across subjects, we also confirmed a significant increase (ranging from 40% to 70%) in cortical region participation at the subject level in multi-area spindles with respect to the local spindles (*Figure 2—figure supplement 1a*). We then compared spindles detected by the CNN and AT approaches. To do this, we first computed the ratio of spindles detected by the CNN and AT for all classes. This comparison revealed that multi-area spindles were systematically detected approximately 1.5 (ECoG) to 10 (iEEG) times more often with the CNN than with the AT (*Figure 2c* and *Figure 2—figure supplement 1b*). Across all recordings, the increase in the multi-area spindles detected by the CNN was significantly greater than in the local spindles (p<1 × 10⁻³, NHP ECoG recordings; p<1 × 10⁻⁵, EEG recordings; p<0.02, iEEG recordings, one-sided Wilcoxon signed-rank test; similar results for the local-regional comparison, p<0.02, EEG recordings; p<0.01, iEEG recordings, one-sided Wilcoxon signed-rank test, n.s. in NHP ECoG). Importantly, iEEG has the highest spatial resolution across the recording types studied here (*Mukamel and Fried, 2012*) and also exhibits the largest increase in multi-area spindles detected by the CNN versus the AT. It is important to note, as well, that in the ECoG dataset the CNN approach detects fewer local and regional spindles than the AT (*Figure 2c*). This effect was primarily due to estimated threshold varying widely across electrodes in one subject, which in turn caused more detections of local and regional spindles in the AT. Next, we computed spindle participation at the level of cortical lobes (frontal, temporal, parietal, occipital) and cortical systems (executive, limbic, visual, auditory, somatosensory) and detected a significant increase in multi-lobe and multi-system spindles across all recordings (*Figure 2—figure supplement 2*). Taken together, these results demonstrate that spindles appear much more widespread across cortex when detected using our approximately amplitude-invariant deep learning approach.

The organization of spindles across the cortex is thus neither fully local nor fully global: the co-occurrence patterns of this sleep rhythm contain a mixture of local and widespread events. If this is the case, how can pre-sleep memory engagement impact this distribution? To answer this question, we further studied the human EEG dataset, which had the unique feature of testing sleep after tasks with varying memory loads. Briefly, before nap EEG recordings, subjects completed a task in which five

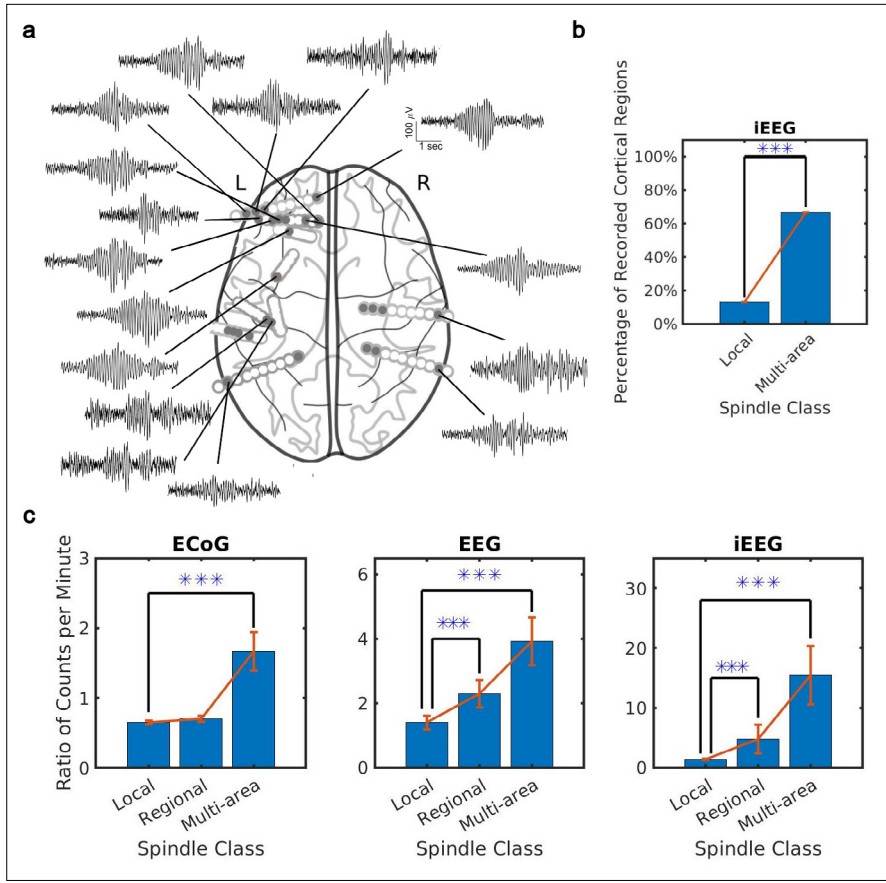

**Figure 2.** Distribution of the extent of spindles detected by convolutional neural network (CNN) and amplitude-thresholding (AT) approaches. (**a**) An example of a widespread, multi-area spindle with electrode sites distributed widely across the cortex. Filled gray circles indicate electrode contacts in gray matter. (**b**) Plotted is the percentage of unique recorded cortical regions with spindles detected by the CNN in the local versus multi-area case across all subjects in the intracranial electroencephalogram (iEEG) recordings (average ± SEM; n = 389445 for local, n = 28407 for multi-area; p < 1 × 10$^{-10}$, iEEG recordings, local versus multi-area, one-sided Wilcoxon signed-rank test). Results were similar at the level of individual subjects (*Figure 2—figure supplement 1a*). (**c**) Plotted are the ratios of spindles detected by the CNN and AT in non-human primate (NHP) electrocorticography (ECoG) recordings (left, n = 13), human electroencephalogram (EEG) recordings (middle, n = 32), and iEEG recordings (right, n = 89) in local (1–2 sites), regional (3–10 sites), and multi-area (more than 10 sites) spindle classes (average ± SEM in all cases; p > 0.1, NHP ECoG recordings; p < 0.02, EEG recordings; p < 0.01, iEEG recordings, local versus regional comparison, one-sided Wilcoxon signed-rank test; p < 1 x 10$^{-3}$, NHP ECoG recordings; p < 1 x 10$^{-5}$, EEG recordings; p < 0.02, iEEG recordings, local versus multi-area comparison, one-sided Wilcoxon signed-rank test). Across recordings, the increase in regional and multi-area spindles detected by the CNN is significantly larger than for the local spindles (except local versus regional in the NHP ECoG).

The online version of this article includes the following figure supplement(s) for figure 2:

**Figure supplement 1.** Extent of spindles detected by convolutional neural network (CNN) and amplitude-thresholding (AT) approaches.

**Figure supplement 2.** Extent of spindles detected by convolutional neural network (CNN) and amplitude-thresholding (AT) approaches across cortical lobes and systems.

novel outdoor scenes (H-VM) or two novel outdoor scenes (L-VM) were required to be held in working memory for 6 s (*Figure 3a*). After the delay period, subjects were then presented with a subsequent visual scene and asked whether it belonged to the previously presented set. In each case (H-VM and L-VM), trials were balanced so that the same total number of visual scenes was presented before sleep. An increase in spindle density after memory tasks and its relationship with memory consolidation is well established (*Clemens et al., 2005*; *Dang-Vu et al., 2008*; *Gais et al., 2002*; *Schabus et al., 2007*; *Schabus et al., 2004*); however, the effect of memory tasks on co-occurrence remains

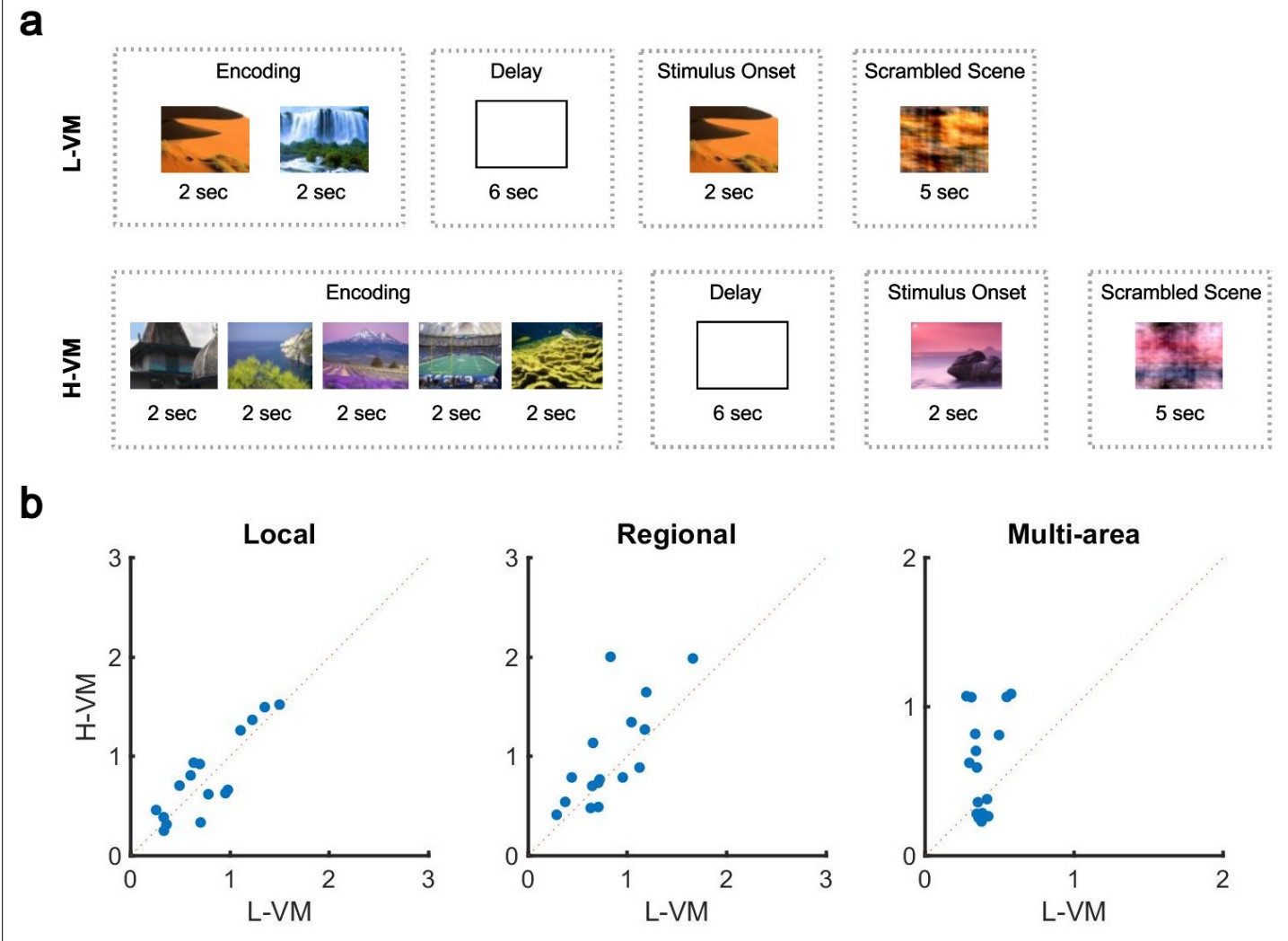

**Figure 3.** Impact of visual memory load on multi-electrode sleep spindle occurrence. (**a**) Schematic representation of low- and high-load visual memory tasks. (**b**) Multi-electrode spindle rate (average number of spindles detected per minute across the array) in high versus low visual memory condition. Spindles are grouped into local (left), regional (middle), and multi-area (right) classes as detected by the convolutional neural network (CNN) model. A significant increase in the number of spindles among subjects can be observed in multi-area and regional spindles as opposed to local spindles ($p>0.34$, local spindles; $p<0.038$, regional spindles; $p<0.02$, multi-area spindles; one-sided paired-sample Wilcoxon signed-rank test).

The online version of this article includes the following figure supplement(s) for figure 3:

**Figure supplement 1.** Low- and high-load visual memory task and its impact on sleep spindle occurrence – signal-to-noise ratio (SNR) and amplitude-thresholding (AT).

**Figure supplement 2.** Cortical lobe participation.

unknown. Considering the potential circuit mechanism for spindles to link activity in neuron groups distributed across multiple areas in cortex through long-range excitatory connections (*Muller et al., 2016*), we then hypothesized that sleep following H-VM tasks would exhibit more multi-area spindles and a larger spatial extent. To test this hypothesis, we first confirmed that amplitudes of detected spindles did not differ across L-VM and H-VM conditions ($p>0.77$, Wilcoxon signed-rank test). We then defined a 'multi-electrode' spindle rate, which considers spindles occurring simultaneously on several electrodes as a single event. Importantly, this multi-electrode spindle rate is distinct from the 'single-electrode' spindle rate computed previously, where spindles occurring simultaneously across multiple electrodes are not recognized as the same event. We next computed the multi-electrode rate for local, regional, and multi-area spindles after L-VM and H-VM tasks. Both regional and multi-area spindles appeared more often after H-VM than L-VM ($p<0.038$, regional spindles; $p<0.026$, multi-area spindles;

one-sided paired-sample Wilcoxon signed-rank test; average ± SEM multi-electrode rates 0.77±0.10 [0.79±0.11], 0.82±0.09 [0.99±0.13], and 0.38±0.02 [0.62±0.08] for local, regional, and multi-area spindles, respectively, in L-VM [H-VM in square brackets]) as detected by the CNN model, consistent with our hypothesis (*Figure 3b*), while local spindles did not appear more frequently (p>0.34, same test). Similarly, the largest increases following H-VM versus L-VM were observed in the subset of multi-area spindles detected by the more-conservative SNR approach (*Figure 3—figure supplement 1a*); however, no increase in multi-area spindles was observed with the AT algorithm (*Figure 3—figure supplement 1b*). The CNN model and SNR approach thus provide clearly converging evidence that an increase in distributed spindles appears following H-VM tasks, a change that is not detected by the AT approach. These results not only validate the performance of the CNN approach in contrast to amplitude-based approaches, but also clearly demonstrate that this approach is able to find qualitatively new results providing insight into the process of human memory consolidation. Lastly, we divided EEG electrodes based on their cortical lobe (*Figure 3—figure supplement 2a*) and studied the change in density of spindles in frontal, occipital, and parietal lobes in low and high visual memory conditions. To do this, in each cortical lobe, we computed the percentage of electrode sites with spindles within the detected windows by the CNN model. Interestingly, we observed a significant increase in the electrode participation during spindles in H-VM versus L-VM across cortical lobes, with the largest increase in the occipital lobe and lowest in the frontal lobe (*Figure 3—figure supplement 2b*).

We then studied the spindles detected by our CNN approach in the EEG dataset further, by applying techniques previously developed to study the spatiotemporal organization of spindles across electrodes during individual oscillation cycles (*Muller et al., 2016*; *Muller et al., 2014*). We first computed the average organization into rotating waves traveling from temporal, to parietal, and on to frontal lobe (denoted 'TPF waves') and waves rotating in the opposite direction (first temporal, then frontal, and then parietal lobe, denoted 'TFP waves'), over all spindle events in H-VM and L-VM conditions. We observed a significant shift toward TFP waves in the H-VM versus L-VM condition (p>0.50, TPF direction; p<0.003, TFP direction; one-sided paired-sample Wilcoxon signed-rank test) (*Figure 4a*). This increase in TFP waves under conditions of high memory load disappears when we restrict the analysis to local spindles (*Figure 4b*), consistent with the idea that these TFP waves may be related to multi-area spindles (p>0.42, TPF direction; p>0.63, TFP direction; one-sided paired-sample Wilcoxon signed-rank test). We then repeated the analysis over all electrodes during multi-area spindles and again observed a significant shift in the average TFP rotating waves (p>0.40, TPF direction; p<0.007, TFP direction; one-sided paired-sample Wilcoxon signed-rank test) (*Figure 4c* and *Video 1*). These results demonstrate that multi-area spindles detected by the CNN model exhibit a clear rotating wave pattern which increases in the TFP direction under conditions of high memory load.

## Discussion

These results may tie the increase in multi-area spindles to the neural circuit mechanism that we have previously identified could play a role in consolidating memories across distributed networks in cortex through synaptic plasticity (*Muller et al., 2016*). Synaptic plasticity occurs through spike time-dependent plasticity (STDP) (*Bi and Poo, 1998*; *Markram et al., 1997*), for which presynaptic vesicle release and postsynaptic spiking must occur with a precision of a few milliseconds (*Magee and Johnston, 1997*). While it has become increasingly clear that sleep spindles play an active and causal role in sleep-dependent memory consolidation (*Aton et al., 2014*; *Clemens et al., 2005*; *Eschenko et al., 2006*; *Gais et al., 2002*; *Mednick et al., 2013*; *Rasch and Born, 2013*), it remains unclear how these oscillations coordinate activity across areas to shape neocortical assemblies distributed over long distances (*Klinzing et al., 2019*). In previous work, we studied the spatiotemporal dynamics of the sleep spindle oscillation in intracranial ECoG recordings from human clinical patients, and we found that – instead of being synchronized with zero delay throughout the cortex – sleep spindles are often organized into rotating waves traveling across the cortex in a preferred direction (*Muller et al., 2016*). Because these waves travel at the same speed as axonal conduction across long-range white matter fiber networks in cortex, the offsets of activity across areas could precisely align spikes across areas to create the conditions necessary for bi-directional synaptic plasticity (*Figure 5*) – either for creating strong links between assemblies distributed widely across cortex or for downscaling connections to maintain synaptic homeostasis (*Crunelli et al., 2018*; *Klinzing et al., 2019*; *Tononi and Cirelli, 2014*). Importantly, previous theoretical work has identified the relative phase of sending and receiving

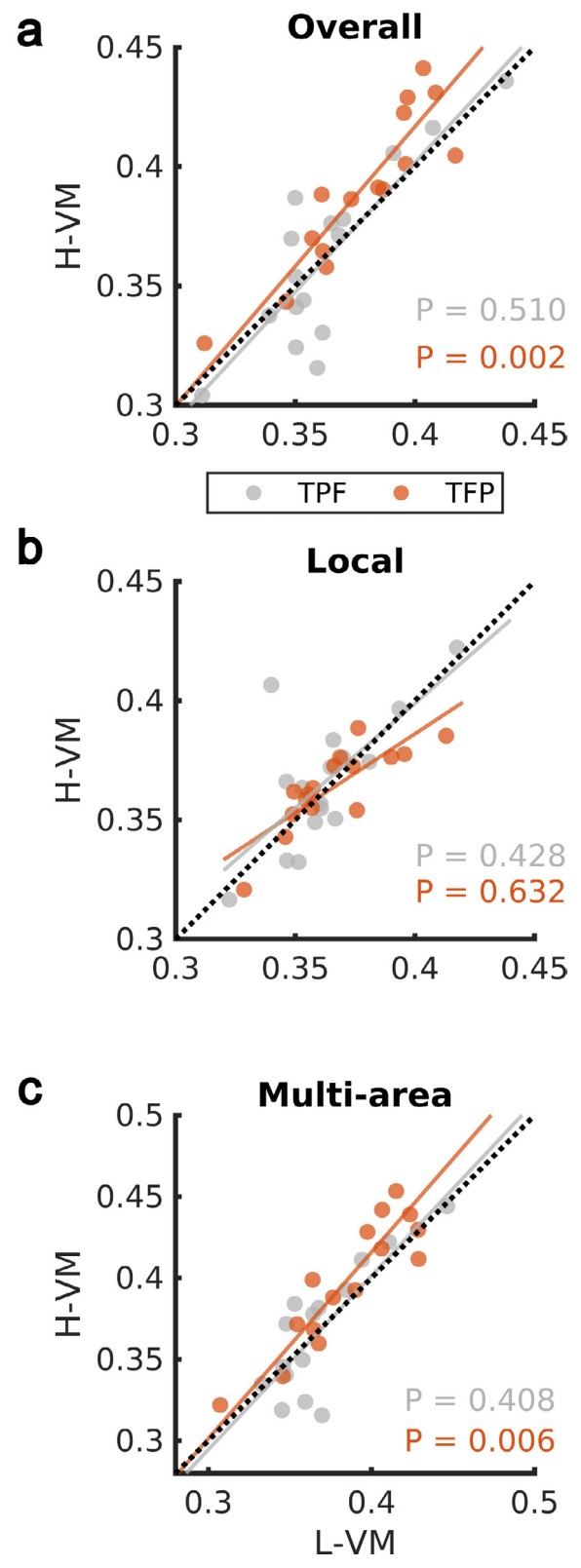

**Figure 4.** Impact of visual memory load on rotating waves. (**a**) Average TPF (gray) and TFP (red) rotation directions computed over all spindle activities detected by the convolutional neural network (CNN) model in high versus low visual memory condition. A significant increase in the TFP direction was observed as opposed to the TPF direction in the high visual memory conditions (one-sided paired-sample Wilcoxon signed-rank test). An outlier point

*Figure 4 continued on next page*

*Figure 4 continued*

(low-load visual memory [L-VM], high-load visual memory [H-VM]): (0.48,0.51) in the TFP direction was omitted for the sake of visualization. (**b**) Average TPF rotation direction (gray) and TFP rotation direction (red) computed over just local spindles. No significant increase was observed in both directions (one-sided paired-sample Wilcoxon signed-rank test). (**c**) Finally, average TPF rotation direction (gray) and TFP rotation direction (red) computed over all electrodes during multi-area spindles. The increase in TFP directions became significant in high visual memory conditions in multi-area spindles (one-sided paired-sample Wilcoxon signed-rank test) which verifies that the increase is driven by the multi-area spindles. An outlier point (L-VM, H-VM): (0.51,0.56) in the TPF direction was omitted for the sake of visualization.

The online version of this article includes the following figure supplement(s) for figure 4:

**Figure supplement 1.** Average rotation direction (surrogate data – simulated rotating waves).

populations as an important factor in determining the balance toward potentiation or depotentiation by STDP during these rhythms (*Muller et al., 2011*) and future computational analyses could study these offsets in detail. At the circuit level, thalamocortical circuits may set the rhythm for spindle oscillations in cortex (*Clascá et al., 2012*; *Destexhe and Sejnowski, 2001*), which are then shaped into waves by long-range corticocortical connections (and their axonal time delays). Understanding the network mechanism for this interplay between thalamocortical and corticocortical connections is thus an important subject for future computational analyses and network models.

This mechanism places the spatial extent of spindles across cortex, and how this extent changes under different memory conditions, as a critical point in understanding the neural process of sleep-dependent memory consolidation. The spatial extent of spindles we reported in this work provides a potential mechanism by which long-range excitatory connections between distant populations in cortex could be strengthened during memory consolidation in sleep. Based on this mechanism, we then hypothesized that large, multi-area spindles may exhibit an increase following H-VM tasks. Consistent with this additional hypothesis, both the CNN and SNR methods detect an increase in multi-area spindles and rotating waves uniquely following H-VM tasks. This increase in multi-area spindles was further associated with an increase in waves traveling in the temporal → frontal → parietal (TFP) direction. Interestingly, these TFP waves are opposite to the dominant rotation direction observed in previous work (*Muller et al., 2016*), potentially reflecting increased top-down influence from higher cortical areas following the high-load memory condition. These present results clearly indicate that different memory conditions can modulate the extent and spatiotemporal organization of sleep spindles across cortex; however, future analyses of intracranial recordings at very high spatial and temporal resolution during memory tasks will be needed to fully understand the spatiotemporal dynamics reported here and their connection to the process of sleep-dependent memory consolidation.

Previous work has found that spindles can occur broadly across the cortex with low measures of synchrony in EEG and magnetoencephalography (*Dehghani et al., 2011a*; *Dehghani et al., 2011b*, *Dehghani et al., 2010*), and also in iEEG recordings (*Frauscher et al., 2015*). The results reported in our work may provide insight into the underlying mechanism for these previous findings: because traveling waves introduce systematic phase offsets across sites, traveling waves during multi-area spindles can account for increases in spindle power across broad regions of cortex that also show low synchrony (as measured, for example, by correlation between electrode sites in *Frauscher et al., 2015*). The results reported in this work highlight the importance of distinguishing between the extent of spindle occurrence (as with

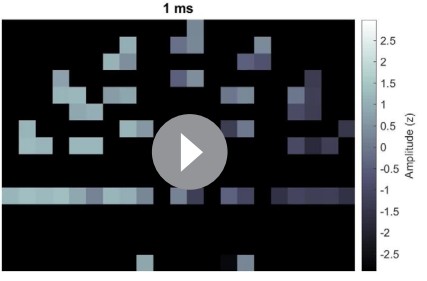

**Video 1.** Rotating waves in multi-area spindles. An example of a rotating wave in TFP direction during a multi-area spindle detected by the convolutional neural network (CNN) model in the electroencephalogram (EEG) recording. Z-score of bandpass filtered (here 9–18 Hz) signals are plotted in falsecolor in a lateral view of the scalp EEG (where frontal, temporal, and parietal lobes are, respectively, located on the right-hand side, the bottom center and top center).
https://elifesciences.org/articles/75769/figures#video1

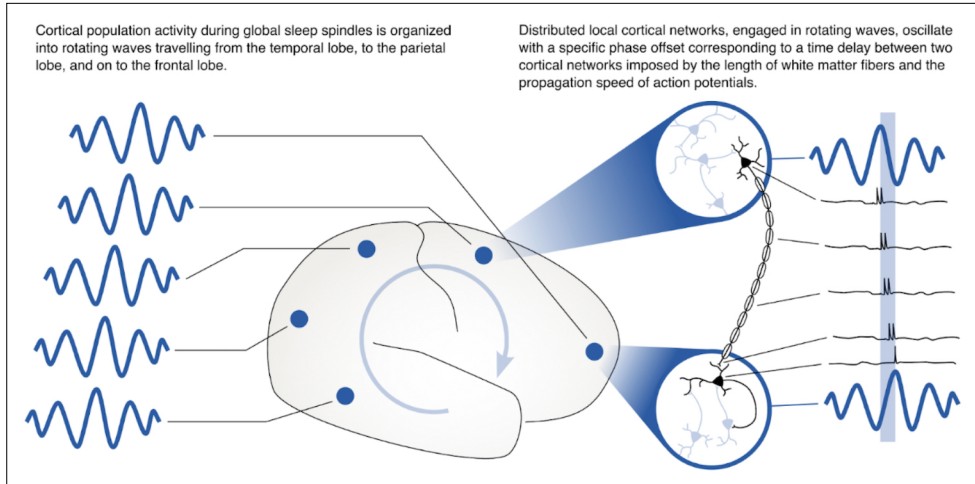

Cortical population activity during global sleep spindles is organized into rotating waves travelling from the temporal lobe, to the parietal lobe, and on to the frontal lobe.

Distributed local cortical networks, engaged in rotating waves, oscillate with a specific phase offset corresponding to a time delay between two cortical networks imposed by the length of white matter fibers and the propagation speed of action potentials.

**Figure 5.** Rotating waves during multi-area sleep spindles provide a mechanism for linking local neuronal populations distributed across cortex. (Left) Spindles that appear across multiple areas are often organized into rotating waves in human cortex. (Right) Phase offsets between cortical regions emerging during rotating waves correspond to axonal conduction delays of white matter fibers and can provide a mechanism to align spikes between cell populations distributed widely across cortex.

detecting individual spindle events in this work) and measures of synchrony. Further comparisons focusing on spatiotemporal dynamics and spindle synchrony, for example with simultaneous EEG and iEEG recordings, will be important for future work.

Taken together, these results provide considerable and convergent evidence from both human and NHP sleep that (1) the spatial extent of sleep spindles was previously underestimated by AT approaches (which tend to select only the highest-amplitude events) (*Figure 2*), (2) this spatial extent can be modulated by the specific memory conditions prior to sleep (*Figure 3*), and (3) increased spatial extent in sleep spindles following H-VM tasks is also associated with rotating waves traveling in a specific direction across cortex (*Figure 4*). To analyze these sleep recordings, we adapted newly developed deep learning approaches for detecting rhythmic events in high-noise data (*George and Huerta, 2018*; *Perol et al., 2018*). The detection process involves two steps: first, we use a simple algorithm (here, the SNR approach) to detect a subset of high-quality examples that can be used for training the CNN, and second, we use the CNN to detect events throughout the recording. The fact that this two-step training approach works well on recordings with very different electrode types and spatial sampling (ranging from scalp EEG to invasive intracranial depth electrodes, *Figure 1—figure supplement 3a*) demonstrates promise of this computational tool for analysis of other rhythmic waveforms that may be of interest in high-noise biological recordings.

We believe this two-step approach represents a methodological advance, coupling a constrained initial detection step with a CNN model that can detect a comprehensive set of events in noisy neural recordings. Our toolbox for this two-step training protocol is available online (http://github.com/mullerlab/spindlecnn), with detailed documentation for applications to new neural rhythms and general timeseries data across biology. We believe this technique can provide a first step in addressing an important methodological consideration in analysis of sleep: how can we make population-level statements about a set of neural events detected algorithmically in the absence of a ground truth, without relying on arbitrarily defined thresholds? This technique could be useful throughout biology where questions such as this arise. At the same time, however, we must emphasize that, while the CNN provided a robust method to sensitively detect a comprehensive set of spindles in the sleep recordings studied here, care must be taken both to understand the mechanisms underlying the selections by deep learning algorithms in each case and to validate results on well-constructed controls. Importantly, future work to expand these methods to new sleep phenomena or other biological rhythms should carefully consider the control analyses developed here (e.g., *Figure 1—figure supplements 2 and 10*), along with the analyses developed to understand the features selected by the convolutional model after training (*Figure 1—figure supplements 7 and 8*) and the choice of CNN architecture

(*Figure 1—figure supplement 9*). In particular, the control analyses using the subset of spindles detected by the SNR algorithm represents a useful strategy to validate findings from the CNN, as the SNR method is both highly interpretable and has well-controllable statistical performance (as inspired from the constant false alarm rate technique in radar). We thus believe that this set of methods can inspire future well-controlled studies utilizing open-access data that are increasingly available for computational analysis of neural dynamics in intracranial recordings (*Boran et al., 2020*; *Frauscher et al., 2018*; *Nejedly et al., 2020*; *Stolk et al., 2018*; *von Ellenrieder et al., 2020*).

## Materials and methods

### Recordings

We studied performance of the CNN model across three sleep datasets that include full recordings annotated as sleep without excluding REM states obtained from electrodes ranging from traditional scalp EEG to invasive intracranial depth electrodes. These datasets represent recordings from very different electrode types, which vary widely in resolution and SNR. Training the CNN model in the same way over these very different recordings demonstrates the generality of the framework developed here; further, these results also represent a cross-species comparison of sleep-rhythm dynamics in NHP and human neocortex.

The first dataset contains ECoG recording from most of the lateral cortex in two macaques during natural sleeping conditions (*Yanagawa et al., 2013*). Recordings were obtained from 128 electrodes in both monkeys and sampled at 1 kHz by a Cerebus data acquisition system (Blackrock Microsystems, Salt Lake City, UT). Sleep state was determined by the degree of spatial synchronization in slow wave oscillations and a significant increase in delta power was reported in sleep condition versus waking activity (*Chauvette et al., 2011*; *Dang-Vu et al., 2005*; *Destexhe et al., 1999*; *Murphy et al., 2009*). This dataset was recorded and distributed by Laboratory for Adaptive Intelligence, BSI, RIKEN, and was made freely available at http://neurotycho.org/anesthesia-and-sleep-task.

The second dataset contains high-density scalp EEG recording from 20 healthy participants (*Mei et al., 2018*). Each participant participated in two separate sessions and completed an H-VM and L-VM task. The recordings were obtained during naps following the visual memory tasks from a 64-electrode EEG skull cap and sampled at 1 kHz. Sleep state was manually assessed by an expert for stage 2 NREM sleep. Ultimately, sleep recordings that did not reach stage 2 sleep or were too noisy were excluded from the study. Under these criteria, four subjects were excluded (subjects 12, 20, 26, and 27). In addition, the recordings were common average referenced to remove large artifacts with potentially non-neural origin. These recordings were made freely available at the Open Science Framework through the link https://osf.io/chav7.

The last dataset contains iEEG recordings from five epileptic patients in the Epilepsy Monitoring Unit (EMU) at London Health Sciences Centre (LHSC). Patients were implanted using depth electrodes for the sole purpose of surgical evaluation. Informed consent was collected from the patients in accordance with local Research Ethics Board (REB) guidelines. Each patient was implanted with 9–15 iEEG electrodes located across the cortex with up to 10 contacts in gray or white matter. The iEEG signals were recorded continuously for a duration of 7–14 days for the purpose of seizure localization. We used clinically annotated sleep onsets and studied half an hour recording starting from the beginning of the sleep/nap cycles in electrode contacts located within gray matter.

### SNR measure for sleep spindle detection

To specify a subset of spindles required to train our CNN model, we implemented a modified version of SNR algorithm (*Muller et al., 2016*). This algorithm, which is inspired by the adaptive, CFAR technique in radar, was used to detect narrow-band rhythmic activities. We measure the ratio of power within the frequency band of interest (here, 9–18 Hz) to power in the rest of the spectrum (1–100 Hz bandpass, with band-stop at 9–18 Hz) at each electrode. The SNR measure is computed over a sliding window of time (500 ms) and produces an estimate of how power in the frequency band of interest compares to total power in the recording, taking into account the noise on individual electrodes. We then used the SNR algorithm to produce high-quality training samples for the CNN model. To do this, we reduced the probability of false positives by setting the threshold to the 99th percentile of the SNR distribution, thus detecting only the activity patterns that have the highest unique power

concentration in the spindle frequency range. We additionally required the SNR algorithm to only include activities with a duration between 0.5 and 3 s, consistent with the duration of sleep spindles. The detected windows are then used for training the CNN model.

To additionally verify performance of the SNR algorithm, we implemented this approach over 1 s recordings of a 90 by 90 array of local field potentials (LFPs) generated by a spiking network model of cortical activity in the awake state (*Davis et al., 2021*), which does not contain the thalamic reticular loops and thalamocortical projections needed to generate sleep spindles. This model, composed of several million neurons with biologically realistic synaptic connectivity (and several thousand synapses per cell), creates realistic ongoing activity patterns consistent with the well-studied asynchronous-irregular state, corresponding to activity in the cortex of awake animals (*Destexhe, 2009*). In addition, this model utilizes a recently developed LFP proxy (*Mazzoni et al., 2015*), allowing us to analyze a population signal using our spindle detection algorithm. SNR values calculated from these data were uniformly below 0 dB, confirming the robustness of our approach in uniquely detecting spindle frequency activity through a known ground truth dataset.

## CNN for sleep spindle detection

We developed a CNN to detect spindles during sleep. The model is motivated by the successful implementation of convolutional networks for waveform detection with earthquakes and gravitational waves in high-noise settings (*George and Huerta, 2018*; *Perol et al., 2018*). If trained properly, it has the ability to detect clearly formed spindles ranging from low to high amplitudes (*Figure 1d*) and provides a great opportunity to study the spatial and temporal analysis of spindle activities across the cortex. We implemented an architecture similar to the one proposed by *George and Huerta, 2018*, with small modifications to the input and convolutional layers to take into account the basic features of the spindle rhythm in cortex (e.g. average duration). The CNN architecture is also slightly tailored to different sampling rates in each recording modality. As in previous work, the convolutional layer is designed to start by extracting local features, gradually extracting longer-timescale features by decreasing the feature space. Using this strategy, the CNN model can efficiently learn to detect the specific waveform characteristics of the sleep spindle in different types of recordings.

We verified model quality using ECoG recordings by minimizing the difference between predicted and training labels marked by the SNR approach. In addition, we verified that the proposed CNN model is not sensitive to the slight change in the number of layers (e.g. 4, 5, and 6 convolutional layers) and hyperparameters such as learning rate, maximum number of epochs used for training, and pooling parameters by conducting a comprehensive sensitivity analysis (*Supplementary file 3*). To perform this sensitivity analysis, for each CNN architecture, we made a grid search over the potential range of hyperparameters, measuring the similarity of model output by Cronbach's alpha. Similarity across hyperparameters within 10–50% of those used in our analysis was greater than 0.96, indicating high reproducibility under moderate parameter variability. We selected one of the best architectures and a combination of hyperparameters that we tested in the grid search (*Figure 1c*). We then used the same architecture and hyperparameters across all subjects and recording datasets. We trained a separate CNN model for each subject on a portion of the available recording and then applied the trained model to detect spindles across the entire recording. Our CNN model is a one-dimensional (1D) model (applied always independently to individual electrodes in the recording) with five convolutional layers (with 32, 64, 128, 192, and 256 filters) and four fully connected layers (with sizes 128, 64, 32, and 2). Each convolutional layer is followed by a maxpool and rectified linear unit layers, and the output of the fifth convolutional layer is gradually flattened into 2D vectors using the fully connected layers followed by rectified linear unit layers. Our classifier has an additional softmax layer at the end which returns the probability of a spindle in addition to the predicted label.

We trained a separate CNN model for each subject over a subset of spindle and non-spindle windows selected from the sleep recordings. To deal with the classification problem of a highly imbalanced training set, we randomly selected a subset of non-spindle windows (up to twice the number of spindles) and then trained the model over the new dataset. In our training process, we noticed that a subset of approximately 1500 windows of spindle and non-spindles can provide enough data for training the CNN model. After training the CNN model, we implemented the model over the entire sleep recording. The CNN model takes a sliding window of sleep recording (500 ms which is bandpass filtered at 1–100 Hz after removal of line noise and harmonics) as an input and predicts its

label (spindle or non-spindle). The sliding window starts at the beginning of the recording and moves 100 ms in each step. To find the start and end time of a spindle, we first combined all overlapping spindle windows. We then included neighbor windows if there is any spindle within 100 ms of the combined windows to account for potentially mislabeled windows. The start of a spindle is finally set to the beginning of the first window, and the end of the spindle is set to the end of the last window. To study the spatial extent of spindles, we then classify detected events as local (1–2 recording sites), regional (3–10 sites), or multi-area (more than 10 sites) in each window. We note that classifying events in this way allows correctly characterizing an event that starts as a local spindle and then evolves to a multi-area event, by counting the times where the event was local separately from the times where the event was distributed across many sites.

### PSD estimate

To verify performance of the CNN, SNR, and AT approaches, we compared PSD estimates of spindle and non-spindle activities (Welch's method; *Figure 1b*, *Figure 1—figure supplements 3b and 10*). In both cases, we first removed line noise artifacts. We then computed PSD over windows of 0.5 s with no overlap and average spectra over detected events. Matched non-spindle PSDs were estimated over a large number of randomly selected non-spindle windows. The increase in the power during the natural frequency range of sleep (~9–18 Hz) in spindle versus non-spindles activities demonstrates the ability of both the CNN model and SNR algorithm to correctly identify spindle activities.

### Time-shifted averaging control

As an additional control analysis, we computed averages over detected spindles, with activity shifted to centrally align the largest oscillation peak in the detected time window. To compute this average, we first needed to correct for the time offset between different spindles. To do this, we shifted detected spindles to the largest positive value within the detected window, corresponding to the positive potential of an individual spindle oscillation cycle, and then took the average over all time-shifted windows. The average of time-shifted signals is computed over spindle windows detected by the CNN approach, as well as matched randomly selected non-spindle windows. Importantly, while the time-shifted average clearly exhibits 11–15 Hz oscillatory structure when computed over spindle events detected by the CNN, this need not be the case, as demonstrated by application of the same approach to matched non-spindle events (*Figure 1—figure supplement 2*). The peaks observed in the center of the signal averaged over non-spindle windows are due to the alignment procedure. Naturally, this peak exhibited a decay consistent with the autocorrelation time present in the signal; importantly, however, it shows no oscillatory structure consistent with spindle activity in the non-spindle windows. This result demonstrates that our CNN model can correctly distinguish between spindle and non-spindle events.

We also systematically studied the sensitivity of the CNN model as a function of the SNR threshold used for building the training set. To do this, we computed the time-shifted average over spindle events detected by the CNN model at different levels of the SNR threshold (*Figure 1—figure supplement 11*). Clear, well-formed 11–15 Hz oscillatory structure is observed in the time-shifted averages above 0 dB threshold, verifying the quality of detected spindles by the corresponding CNN models. However, the 11–15 Hz oscillatory structure starts to disappear below 0 dB because an SNR threshold below 0 dB introduces errors into the training sets by mislabeling noise signals as spindles. On the other hand, similar oscillatory shapes of time-shifted average above 0 dB confirms the ability of the CNN model to perform robustly while trained over different sets of clearly formed spindles.

### Comparison with AT approach

The AT approach has been used extensively in the literature to automatically detect spindles during sleep (*Gais et al., 2002*; *Nir et al., 2011*). In this approach, a spindle is detected when the amplitude of the bandpass signal stays above a threshold for a limited period of time (e.g. at least 500 ms; cf. Figure S5 in *Nir et al., 2011*). To implement this approach, we first bandpass filter the signal at the frequency of 11–15 Hz and then compute the signal envelope using Hilbert transformation over a sliding window of 0.5 s. The sliding window starts at the beginning of the recording and moves 100 ms in each step. The start and end time of each spindle is computed similarly as with the CNN model,

where we combine overlapping spindle windows and neighbor windows within 100 ms. A spindle is detected whenever the signal envelope stays above the predetermined threshold for at most 3 s.

To determine the most appropriate threshold for comparison to the CNN and SNR approaches, we first computed the distribution of electrode-level RMS amplitude that results in approximately 2 spindles per minute and then set the overall threshold to its average across all electrodes. The overall threshold is computed independently for each subject to account for the differences across subjects as well as across different electrodes. The quality and extent of detected spindles by the AT approach was then compared with the CNN and SNR (*Figure 2c*, *Figure 1—figure supplements 1 and 3*, 4, 10, and 12, *Figure 2—figure supplements 1 and 2*, and *Figure 3—figure supplement 1*). The CNN model has a relatively amplitude-invariant nature in comparison with the AT approach, which is highly sensitive to a predefined cutoff AT. The AT approaches may only select spindles with the largest-amplitude events, or could miss ones that temporarily dip below the threshold, while our approach has the ability to find well-formed spindles that are both large and small in amplitude (cf. first and third EEG spindles of *Figure 1d* which were not detected by AT and *Figure 1—figure supplement 4*). Consistent with our expectation, the AT approach detects spindles of higher amplitude than the CNN approach (*Figure 1—figure supplement 3b*) with the exception of the EEG dataset, where lower SNR may obscure this effect.

## Simulated data control – signal amplitude

We simulated 60 min of recording containing on average 3 spindles per minute. The spindles were simulated using

$$f(t) = Ae^{i(\omega t + \theta_0)}, \tag{1}$$

where $A$ is the oscillation amplitude, $\omega$ is the oscillation angular frequency, $t \in [1, N_t]$ is sample number, $\theta_0$ is the initial polar angle. Oscillation amplitude ($A$) was set to a constant value, and the oscillation angular frequency ($\omega$) and initial polar angle ($\theta_0$) were randomly selected from, respectively, 11–15 Hz spindle frequency range and $[0, 2\pi]$. We also added two types of noise to the signal including white noise with constant power spectrum, and Brownian noise with $1/f^2$ power spectrum.

We utilized these surrogate data, which have a clear PSD peak in the spindle frequency range (*Figure 1—figure supplement 1a*), to study how sensitive the SNR approach is to the change in spindle and noise amplitude as opposed to the AT techniques. We first applied the SNR algorithm and AT approach to the simulated signal to detect spindle activities. For the SNR approach, we used either the 99th percentile of the SNR distribution, or 0 dB if greater (which represents parity between power in the spindle passband and the rest of the signal spectrum) as the threshold. For the AT approach, we used three standard deviations of instantaneous amplitude. We then repeated the entire analysis once after we doubled the noise amplitude, and once after dividing the spindle amplitude in half. *Figure 1—figure supplement 1b* contains an example of spindle activity detected by both approaches in the original signal. Interestingly, in the higher-noise setting, as well as with lower-amplitude spindles, the SNR approach was still able to detect the spindle activity while the AT approach failed to detect the spindles (*Figure 1—figure supplement 1c and d*). These results demonstrate the superior and robust performance of the SNR algorithm in face of changes in signal and noise amplitude.

## Simulated data control – varying noise

We simulated 30 min recording of a 10 by 10 array of electrodes containing on average 2 spindles per minute. The spindles were simulated using *Equation 1* where oscillation amplitude ($A$) was set such that it follows the standard spindle waning and waxing pattern, and the oscillation angular frequency ($\omega$) and initial polar angle ($\theta_0$) were randomly selected from 11 to 15 Hz spindle frequency range and $[0, 2\pi]$, respectively.

We added different types of noise to the signal to verify that the CNN model is relatively general to the type of noise, provided there is enough training data and high-quality marked events. To do this, we chose noise with different type of power spectrum including (1) white noise with constant power spectrum; (2) noise with $1/\sqrt{f}$ power spectrum; (3) pink noise with $1/f$ power spectrum; and lastly (4) Brownian noise with $1/f^2$ power spectrum. We also studied change in the performance of the CNN and AT approaches under biological forms of noise such as REM theta oscillation and non-biological artifacts. We simulated theta oscillations with similar characteristics to the spindle oscillation with

angular frequency ($\omega$) and duration randomly selected from 4 to 8 Hz and 400–1000 ms. For non-biological artifacts, we first randomly chose a subset of artifacts detected as spindles by the AT approach in the iEEG recording. We then used the fast Fourier transform (FFT) to convert these artifacts into frequency domain. We next used the signal amplitude and randomly selected phases from $[0,\ 2\pi]$, to generate a new set of artifacts and used the inverse FFT to convert the signal back to time domain. We visually inspected the simulated artifacts and verified the signals by comparing the PSD of the simulated artifacts with the original artifacts (*Figure 1—figure supplement 6a*).

For each type of noise, we first used the SNR algorithm to generate a subset of high-quality spindles for training the CNN model. After training the CNN model, we implemented the CNN model and AT approach to detect spindle activities. Lastly, we compared the detected activity by CNN, SNR, and AT with the actual spindles (*Supplementary file 1*). In all cases, the CNN was able to efficiently learn to detect the specific waveform characteristics distinguishing the sleep spindle rhythm in these recordings (e.g. *Figure 1—figure supplement 5*) showing that we can expect this approach to generalize well across recordings with different types of noise.

Moreover, we further studied the effect of artifacts on performance of the CNN and AT approach by systematically increasing the number of artifacts per minute in the surrogate data. In particular, the AT approach seems to be very sensitive to recording artifacts. The CNN model performs robustly in face of increases in the number of artifacts, while the performance of the AT gradually decreases as the number of artifacts per minute increases (*Figure 1—figure supplement 6b*). This result further verifies that our approach is not sensitive to different types of artifact in the recording as opposed to the AT approach.

## Rotating wave direction

To estimate the degree of rotational activity in the multi-area spindles, we compute

$$\alpha_{TPF}\left(t\right)\ =\ \left|\left(1/n\right)\sum_{n=1}^{N}\left(e^{ix(n,t)}e^{i\theta(n)}\right)\right|;\qquad(2)$$

$$\alpha_{TFP}\left(t\right)\ =\ \left|\left(1/n\right)\sum_{n=1}^{N}\left(e^{-ix(n,t)}e^{i\theta(n)}\right)\right|;\qquad(3)$$

where $\alpha_{TPF}\left(t\right)$ and $\alpha_{TFP}\left(t\right)$ are, respectively, positive and negative rotation direction at time $t$, $x_t\left(n,t\right)$ is the phase angle of $nth$ electrode at time $t$, $N$ is the total number of electrodes, and $\theta$ is the electrodes' polar angle with respect to the sagittal plane along the midline of the brain. This metric allows us to quantify the strength of the rotational pattern of activity in TPF and TFP on the array of electrodes during spindle activity (*Figure 4*). The proposed metrics are validated using simulated data (*Figure 4—figure supplement 1* and *Videos 2 and 3*) in the following section.

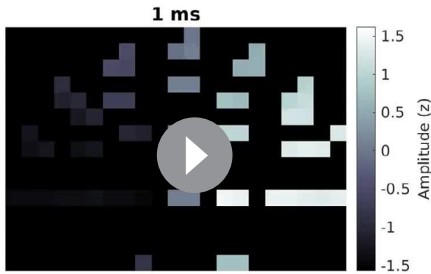

**Video 2.** Simulated TPF waves are well detected by our computational approach. An example of surrogate data, with simulated rotating spindles in the TPF direction. Z-score of bandpass filtered (here 9–18 Hz) signals are plotted in falsecolor in a lateral view of the scalp electroencephalogram (EEG) (where frontal, temporal, and parietal lobes are, respectively, located on the right-hand side, the bottom center and top center).
https://elifesciences.org/articles/75769/figures#video2

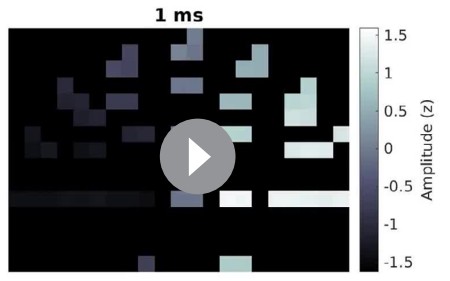

**Video 3.** Simulated TFP waves are well detected by our computational approach. An example surrogate data, with simulated rotating spindles in the TFP direction.
https://elifesciences.org/articles/75769/figures#video3

## Simulated data control – rotating wave

We simulated multi-area spindles with rotational wave organization across the 10–20 system of EEG recordings. We simulated rotating waves using

$$f(t, \theta) = Ae^{i(\omega t - \gamma \theta)} + \sigma \eta(t), \tag{4}$$

where $A$ is the oscillation amplitude, $\omega$ is the oscillation angular frequency, $\gamma$ is the polar wavenumber, $\theta$ is the polar angle with respect to the wave center, $t \in [1, N_t]$ is sample number, and $\eta(t)$ is a real-valued Gaussian white noise term (compare with Equation 12 in *Muller et al., 2016*). Oscillation amplitude ($A$) and polar wave number ($\gamma$) were set to unity, without loss of generality, and oscillation angular frequency, $\omega$, is selected randomly from 11 to 15 Hz spindle frequency range, and lastly, the noise term $\sigma \eta(t)$ was set to zero for the sake of simplicity. We set $\theta$ with respect to the polar angle of electrodes in the 10–20 system and the sagittal plane along the midline of the brain (compare with *Videos 2 and 3*). As expected, the average positive direction estimated over 1000 simulated multi-area spindles was almost 0.9, while the average over the negative direction was almost 0.1, which is close to the case with no synchronized activity (*Figure 4—figure supplement 1a*, *Video 2*). We verified the direction by reversing the traveling wave in our simulation and achieving the opposite result (*Figure 4—figure supplement 1b*, *Video 3*). The simulation results confirm the accuracy of our measure for estimating rotational direction in multi-area spindles.

## CNN visualization and interpretation

To better understand how the CNN model detects spindle oscillation, we studied the filter and saliency maps of the CNN models. To do this, we first simulate six spindle/non-spindle signals (500 ms), including (1) white noise and spindle; (2) white noise and 'half spindle' signal; (3) white noise and combination of two spindles of different amplitude; (4) white noise and theta oscillation; (5) pure white noise; and (6) white noise and Brownian noise. We then visualized the filters and saliency maps across these simulated signals as an input to a CNN model trained on the EEG dataset.

To better visualize the filters, we plotted the output features of the CNN layers. The feature maps visualize CNN filters applied into input signals. In *Figure 1—figure supplement 7*, we plot an example feature map from the last convolutional layer of the CNN model across the six simulated signals. This feature map can accurately detect the maximum amplitude within each cycle using the maximum activation of the CNN model across all oscillations. Interestingly, we observe a relatively similar pattern of activation across the remaining filters, indicating that the timing and relative height of the maximum amplitudes represent the type of activity pattern in the input signal (spindle versus non-spindle). Using these features, the CNN model can learn to reliably detect the specific waveform characteristics of the sleep spindle in different types of recordings.

In addition, we studied gradient attribution maps to identify part of the signal that are most important for classification in the CNN model. To do this, we plot the gradient of the predicted spindle/non-spindle class with respect to simulated signals (*Figure 1—figure supplement 8*). The area of the signal with the highest modulation in amplitude has the greatest impact on the classification. For example, in the 'half spindle' signal, only the half of the signal containing the spindle oscillation is of importance in the classification. In the signal with spindles of different amplitude, the spindle with the highest amplitude is relatively more important than the other half of the signal. The pattern of activations and gradient map across the simulated signals provides insights into the underlying mechanism by which the CNN model efficiently distinguishes between different types of oscillation.

## CNN choice of architecture and hyperparameter setting

The CNN architecture should be tailored with respect to the duration of rhythmic activity, type of oscillation, and sampling rates of recording modality. The duration of the rhythmic activity and sampling rate determine the length of the sliding window for the CNN model. For example, in our NHP EEG recording with the sampling rate of 1000 Hz, the sliding window is set to 0.5 s which is the minimum duration of spindle activities observed during sleep and contains 500 data points. We next specify the filter size with respect to the length of the sliding window and types of rhythmic activity. The CNN layer is designed to start by extracting local features, gradually extracting longer-timescale features by decreasing the feature space. Filter sizes covering approximately one oscillation cycle (70–120 ms)

are effective in detecting spindle activity. To understand this further, we simulate 10 recordings with 2 spindles per minute and add different types of noise and artifacts (*Supplementary file 1*). Using these surrogate recordings for which we have the exact timing of spindles, we demonstrate that longer filter size is ineffective at detecting spindles. Specifically, we gradually increase the filter size (*Figure 1—figure supplement 9*) and compute the performance of the CNN model. As expected, the CNN performance drops as we increase the filter size, verifying this mechanism. This result further validates the generality of the CNN approach for detecting neural rhythms, while also getting at the mechanism. We believe that a similar mechanism can be implemented for specifying the filter size for other neural and biological rhythms. The current CNN architecture works perfectly with slight changes in the sliding window (duration and sampling rate) and type of oscillation, but it requires modification otherwise.

## Electrode localization

For the purpose of electrode localization in the iEEG recordings, we developed an image processing pipeline which involves electrode contact localization, brain tissue segmentation, and atlas fitting. Semi-automatic contact localization was performed in 3D Slicer using the SEEG Assistant (SEEGA) module (*Narizzano et al., 2017*). The entry and target points of each electrode were manually defined on the post-operative CT image. The entry/target labels were provided to the SEEGA algorithm, which automatically segmented the electrode contacts. To obtain brain location information for each contact, brain tissue segmentation and atlas fitting was carried out. To enable the use of anatomical priors during tissue segmentation, the pre-operative T1w MRI was non-linearly registered to the MNI152 2009c Nonlinear Symmetric template (https://www.bic.mni.mcgill.ca/ServicesAtlases/ICBM152NLin2009) using NiftyReg (*Modat et al., 2010*). An anatomical mask was generated by applying the inverse transform to the T1w image using the antsApplyTransforms algorithm from Advanced Normalization Tools 2.2.0 (ANTS; http://stnava.github.io/ANTs; *Cook, 2022*). Segmentation of gray matter, white matter, and cerebrospinal fluid was performed using the Atropos algorithm from ANTS (*Avants et al., 2011b*), which implements $k$-means classification ($k$=3). The resulting posteriors were merged into a 4D volume using the fslmerge algorithm from FMRIB Software Library v6.0 (FSL; https://fsl.fmrib.ox.ac.uk/fsl/fslwiki). The CerebrA atlas (*Manera et al., 2020*) was used to obtain anatomical labels for each electrode contact. Normalization to template space (MNI152NLin2009cAsym) was performed using the non-linear SyN (*Avants et al., 2011a*) symmetric diffeomorphic image registration algorithm from ANTS, using both the brain masks of the pre-operative T1w and template space. Using the inverse of the non-linear transform, the CerebraA atlas labels were warped to the pre-operative T1w MRI space. The atlas labels were then dilated using the fslmaths algorithm from FSL. The final T1w brain tissue/atlas segmentation was mapped to the contacts to provide location information for each contact (tissue probability and brain anatomical region). This custom processing pipeline has been made available on GitHub (https://github.com/akhanf/clinical-atlasreg; *Khan, 2020*).

## Code availability

Our custom MATLAB (MathWorks) implementations of all computational analyses, along with the analysis scripts used for this study are available as an open-access release on GitHub (https://github.com/mullerlab/spindlecnn, swh:1:rev:0c503d103e4a0cf041e43903a896bb25b0c66b9b, *Mofrad, 2022*).

## Acknowledgements

The authors would like to thank the entire team of clinical neurologists and neurosurgeons at London Health Sciences Centre (LHSC, London, Ontario) for their collaboration in this work. The authors would also like to thank the clinical patients for their participation in the research. The authors additionally thank Ingrid Johnsrude, Julio Martinez-Trujillo, and Laura Batterink for helpful comments and discussions on this manuscript. This work was supported by the Canadian Institute for Health Research and NSF (NeuroNex Grant No. 2015276) and BrainsCAN at Western University through the Canada First Research Excellence Fund (CFREF).

## Additional information

### Funding

| Funder | Grant reference number | Author |
|---|---|---|
| Canada First Research Excellence Fund | BrainsCAN at Western University | Lyle Muller |
| National Science Foundation | NeuroNex Grant No. 2015276 | Lyle Muller |
| Fields Institute for Research in Mathematical Sciences | Postdoctoral Fellowship | Maryam H Mofrad |
| Compute Canada | Resource Allocation Competitions (RAC) | Lyle Muller |
| Kyoto University | SPIRITS 2020 | Lyle Muller |

The funders had no role in study design, data collection and interpretation, or the decision to submit the work for publication.

### Author contributions

Maryam H Mofrad, Conceptualization, Data curation, Formal analysis, Methodology, Software, Validation, Visualization, Writing - original draft, Writing - review and editing; Greydon Gilmore, Data curation, Formal analysis, Resources, Software, Validation, Visualization, Writing - review and editing; Dominik Koller, Methodology, Visualization, Writing - review and editing; Seyed M Mirsattari, Jorge G Burneo, David A Steven, Data curation, Investigation, Resources, Supervision, Validation, Writing - review and editing; Ali R Khan, Ana Suller Marti, Data curation, Formal analysis, Methodology, Resources, Software, Supervision, Validation, Writing - review and editing; Lyle Muller, Conceptualization, Formal analysis, Funding acquisition, Investigation, Methodology, Resources, Software, Supervision, Validation, Writing - original draft, Writing - review and editing

### Author ORCIDs

Maryam H Mofrad (ID) http://orcid.org/0000-0002-0094-6694
Dominik Koller (ID) http://orcid.org/0000-0002-7449-9516
Ali R Khan (ID) http://orcid.org/0000-0002-0760-8647
Lyle Muller (ID) http://orcid.org/0000-0001-5165-9890

### Ethics

We analyze a clinical dataset of intracranial electroencephalography (iEEG) recordings from 5 epileptic patients in the Epilepsy Monitoring Unit (EMU) at London Health Sciences Centre (LHSC). Patients were implanted using depth electrodes for the sole purpose of surgical evaluation. Informed consent and consent to publish were collected from the patients in accordance with local Research Ethics Board (REB) guidelines (Western University HSREB #106516 and Lawson REB #114420).

### Decision letter and Author response

Decision letter https://doi.org/10.7554/eLife.75769.sa1
Author response https://doi.org/10.7554/eLife.75769.sa2

## Additional files

### Supplementary files

• Supplementary file 1. Performance of the convolutional neural network (CNN) model and amplitude-thresholding (AT) approach under different types of noise. Examples of CNN and AT implementation using surrogate data with systematically varying noise properties. The CNN models were able to detect the majority of spindles (sensitivity above 97%) subject to different types of noise as opposed to the AT which achieved lower performance (specificity around 72–87%). The CNN model was trained over a subset of high-quality spindles (about 36–41%) detected by the signal-to-noise ratio (SNR) approach using the 99th percentile threshold of the SNR distribution. The high sensitivity and specificity verifies the superior performance of CNN to detect the specific

waveform characteristics distinguishing the sleep spindle rhythm in the recordings with significantly varying types of noise.

• Supplementary file 2. Distribution of gray matter contacts in the cortical regions of all subjects in the intracranial electroencephalogram (iEEG) recordings.

• Supplementary file 3. Convolutional neural network (CNN) hyperparameter sensitivity analysis. Values reported are the average Cronbach's alpha computed over different combinations of CNN hyperparameters which measure the similarity of the outputs (the average alpha computed over the randomly shuffled outputs are approximately 0.005 for each combination). Parameter values in bold are used in the current version of the CNN model.

• Transparent reporting form

### Data availability

Source data for non human primate electrocorticographic (ECoG) recording and human EEG recordings were made freely available through following links: http://neurotycho.org/sleep-task, https://osf.io/chav7. Our custom MATLAB (MathWorks) implementations of all computational analyses, along with the analysis scripts used for this study will be made available as an open-access release on GitHub (http://github.com/mullerlab/spindlecnn, copy archived at swh:1:rev:0c503d103e4a0cf041e43903a896bb25b0c66b9b).

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
