## [Editor Report]

This article provides compelling evidence that deep convolutional networks can detect repeating patterns in biological data better than existing methods, in the presence of noise, biological or otherwise. In analyses of data acquired from the brains of primates using various modalities, the authors show that spindles in cortex have a wider spatial distribution that previously thought. Applications of the proposed approach in other settings may lead to novel findings about the distribution of transient oscillatory patterns in the brain.

---

## [Decision Letter]

**Decision letter after peer review:**

[Editors’ note: the authors submitted for reconsideration following the decision after peer review. What follows is the decision letter after the first round of review.]

Thank you for submitting the paper "Waveform detection by deep learning reveals multi-area spindles that are selectively modulated by memory load" for consideration by *eLife*. Your article has been reviewed by 2 peer reviewers, and the evaluation has been overseen by a Reviewing Editor and a Senior Editor. The following individuals involved in review of your submission have agreed to reveal their identity: Carmen Varela (Reviewer #1).

Comments to the Authors:

We are sorry to say that, after consultation with the reviewers, we have decided that this work will not be considered further for publication by *eLife*.

Overall, as a group, the reviewers found the application of CNNs to the detection of spindles novel and interesting. More generally, the reviewers thought that a method to detect rhythmic behavior from noisy data could have an important impact on neuroscience research. At the same time, the reviewer expressed some reservations, and felt that the manuscript requires additional work to assess the utility of the CNNs as an addition to sleep biologists' toolkits. Should the authors decide to submit a revised version of the manuscript, we will do our best to have it assessed by the reviewing editor and the reviewers who provided the comments below.

The following represent three main critiques/suggestions from the reviews.

1. The reviewers found the comparisons to existing methods for spindle detection too simplistic and lacking. More specifically, the authors compare their method mainly to the AT method. The reviewers found the lack of details of the authors' implementation of the AT method, and the lack of thorough comparison to spindle-detection algorithms more sophisticated than the AT method, such as the SNR method, surprising. The reviewers suggest that the authors describe their implementation of the AT method in more detail (e.g. threshold selection etc…). The reviewers also suggest that the authors perform thorough comparisons of the CNN to methods/algorithms more sophisticated than the AT method, both in simulations and on the data used in the manuscript. Evidence that the CNN detects substantially more spindles than these methods would motivate its adoption by biologists.

2. The reviewers felt that the analyses do not support the authors' interpretations and their claims of new scientific findings. First, differences in spatial sampling of neural activity by EEG, EcOG and iEEG, together with the fact that volume conduction may contribute to synchronous activity, limits the interpretation of the authors result. Second, a line of work by Dehghani, Cash, Halgren and colleagues investigating the spatial extent of spindles suggest a lack of synchrony: e.g. (1) doi: 10.1152/jn.00198.2010, (2) doi: 10.1002/hbm.21183, (3) doi: 10.1016/j.clinph.2010.06.018. The reviewers suggest that the authors temper their claims significantly, and put them in the context of the above literature.

3. The reviewers thought the manuscript could benefit from an expanded discussion of the motivation for utilizing the CNN for spindle detection, in particular of how, if at all, they tailored the architecture to the spindle-detection setting. The description of the architecture (layers etc…) provided in the manuscript seems to mirror the description of that developed for gravitational wave detection. More generally, the reviewers thought that a discussion of how to tailor the architecture to the detection of brain rhythms other than spindles would benefit the neuroscience community. For instance, variables such as the length of the rhythmic pattern of interest ought to impact architectural choices.

4. The reviewing editor asks that, where applicable, the authors modify the manuscript to address the comments from the detailed reviews shared below.

*Reviewer #1:*

Mofrad et al. trained a convolutional neural network (CNN) to improve the detection of multi-area spindles and study their regulation by working memory load. They used the CNN algorithm on three different sleep datasets (scalp EEG from humans that had performed a working memory task, intracranial human EEG, and ECoG recorded from macaques). They find that their CNN method identifies more regional and multi-area sleep spindles compared to an amplitude-based spindle detection method. Importantly, they found that the rate of occurrence of regional and multi-site spindles was higher in the sleep after human subjects performed a high-load visual working memory task, compared to after the subjects performed a low-load visual working memory task.

The results from this study advance the methods used to detect sleep events. As discussed by the authors, most algorithms to detect sleep oscillations in EEG and LFP data are based on amplitude thresholds; this approach is inadequate when studying regional and global spindles, in which the amplitude may not always remain above threshold in all recording sites. The authors overcome this limitation with a ML approach that detects significantly higher numbers of regional and multi-site spindles. The results presented here demonstrate how ML detection algorithms can offer novel insights on the function of spindles; similar approaches could be helpful with other sleep oscillations. Specifically, the results presented by Mofrad et al. suggest that, compared to amplitude-based detection methods, CNNs improve the detection of spindles that occur temporally correlated across cortical regions. In addition, the authors used the CNN spindle detection method on human sleep data after subjects performed a working memory task. The results are consistent with the hypothesis that cross-regional spindle synchronization is regulated by working memory load prior to sleep, providing insight into how specific cognitive processes influence sleep spindles.

An additional strength of this study is the demonstration that the CNN method yields similar results in ECoG, EEG, and iEEG, and in data from macaques and from humans, suggesting that the detection method is applicable across species and across electrophysiological recording methods.

A few points should be considered when interpreting the results. The definition of local, regional, and multi-site spindles used in this manuscript is based on the absolute number of electrode sites in which a spindle was detected. Since the ECoG, EEG, and iEEG have different spatial sampling, the definition could overestimate regional and multi-site spindles in iEEG recordings, particularly in the subjects that had large number of electrode contacts in one brain region. This could contribute to the much higher ratio of global spindles observed in the iEEG data compared to ECoG and EEG. Nonetheless, the analyses that support a correlation between widespread spindles and memory load were based on 64-channel scalp EEG and are not expected to be biased by differences in electrode spatial sampling.

Lastly, the results are interpreted in the context of the systems memory consolidation hypothesis, which proposes that new memories become integrated into neocortical networks in a process that is facilitated by sleep oscillations. The results from the sleep EEG analyses after subjects performed a low- or high- load visual working memory task are indeed consistent with the idea that multi-site spindles contribute to the consolidation of memories formed before the subjects went to sleep. Likewise, the results are also consistent with the synaptic homeostasis hypothesis, which states that sleep oscillations are important in scaling synaptic plasticity of networks that may have been particularly active during wakefulness. The results of this study are therefore congruent with the possibility of multi-site spindles coordinating the homeostatic regulation of synaptic strength across multiple cortical regions in relation to the cognitive load experienced by the subjects during wakefulness prior to sleep.

Recommendations for the authors

The work presented in this manuscript is highly relevant to advance the methods used to detect sleep oscillations, and to understand the functional significance of sleep spindle coordination across cortical areas. The manuscript text and figures are clear and well-organized. Below are suggestions that may help further improve the quality of this work.

The distinction of three spindle types (local, regional, multi-area) based on the co-occurrence across recording sites should be discussed with regard to the different spatial sampling of EEG, ECoG and, specially, iEEG electrodes. For example, spindles detected in multiple recording sites in subjects such as B and C which have multiple contacts in one brain region (Suppl. Table 1), might be incorrectly labelled as regional or multi-area. Although sampling biases are not expected to affect the results regarding memory, it will be helpful for readers to explain how these potential biases were addressed in the analyses of iEEG data. Could the much higher ratio of multi-area spindles in iEEG data (Figure 2c, right panel) be explained by electrode sampling bias? Along the same lines, it may be more appropriate to refer to 'multi-area' spindles as 'multi-site' spindles if they do not necessarily imply occurrence in distinct cortical areas.

It would also be helpful to expand the justification for the choice of CNN algorithms; the text refers to their previous use detecting earthquakes or gravitational waves but is not immediately clear why noise in these applications would be similar to noise in sleep signals.

Line 35: remove 'stage 2': the references provided are mostly from rodents, in which separate NREM sleep stages are less defined; and spindles are not necessarily exclusive of N2 even in humans.

Lines 61,62: Long-range connections are one potential link to temporarily coordinate spindles across cortical areas. Another important set of connections that could coordinate spindles are the thalamocortical projections, mainly those thalamic pathways with widespread projections to superficial layers of cortex (references such as Clasca et al., 2012, Eur J Neurosci 35(10):1524-32. doi: 10.1111/j.1460-9568.2012.08033.x.).

Lines 102-103: please specify how descriptive statistics are reported; are the values reported in parenthesis the mean and standard deviations? Clarify in the methods section how duration was calculated. The methods state that 'The CNN model takes a window of sleep recording (500 ms which is bandpass…)' (Line 303), is that a 500ms moving window? With what step size? What determines the start and end of a spindle in the CNN and AT methods?

Line 161: include p value for local spindles (as done in the figure legend).

Given the unique dataset and methods, and the focus of this work on the locality and globality of spindles, the authors may want to report on potential changes in the density of local spindles specifically over visual and frontal cortical areas. Is it possible that increases in local spindle density are not significant when all regions are considered together but may be significant in certain regions engaged by the visual working memory task? In that sense, it is interesting that the SNR approach detects a (smaller) increase in local spindles with high memory load (Supp. Figure 4a). A more precise analysis of local spindles over relevant cortical areas will not diminish the author's main result on multi-area spindles, but it will provide additional cues on the role of spindle spatial synchronization in sleep and will enhance the value of this interesting work.

Lines 241-242 say that 'the quality of sleep was studied by the degree of spatial synchronization'; it will be helpful to clarify this statement because if the degree of spatial synchronization was used to select the sleep data used for this work, the dataset may include more synchronized activity compared to other datasets. This does not nullify the results, but clarification will be important to ensure replicability. Likewise, the methods used for sleep detection should be described in more detail, was sleep detected in the same way in all datasets?

Line 250: only visual inspection was used to detect stage 2 in the EEG data? clarify if only NREM sleep was used in all datasets used for analysis? (same for the iEEG recordings)

Not clear what is added by the paragraph between Lines 283-288 unless more details are provided. If the model simulates spiking activity in the awake state, is this helpful as a ground truth control for sleep LFP? What type of spiking model? What network architecture?

Paragraph starting in line 290. Was the CNN model trained and optimized for each dataset?

Lines 296-297: 'best' based on what?

Line 350 states that the overall threshold for the AT method was based on the average root mean squared. Was the AT approach developed and applied independently for each data set (EEG, ECoG, iEEG)? It is not clear from the text if the different signal amplitudes in the datasets were considered for AT detection (were the data and RMS normalized?).

Figure Supp. 5: correct typo in the figure legends (spindle)

*Reviewer #2:*

The primary goal of this paper is to provide evidence that sleep rhythms are more spatially extended than previously known, using a deep learning convolutional neural network (CNN) model specifically designed to characterized rhythmic activity that has not previously been applied to sleep data.

Strengths

1. The authors establish that the CNN method proposed can detect spatially extended sleep rhythms and that difference occur when measuring spindles after a low vs high load memory tasks in multiple data sets.

2. The CNN method presented may provide a powerful technique to characterize a range of brain rhythms and difference across tasks or patient populations.

3. The results and figures are clearly presented.

4. The methods are sound, with the exception that more detail on the method and how it uniquely accounts for rhythmic activity is warranted and should be discussed in the Results section of the paper, rather than in the methods.

Weaknesses

1. The results of the paper do not establish that this CNN method is better than prior methods at detecting spatially extend sleep rhythms, or that the methods is better able to distinguish sleep spindles after a low vs high load memory task. As such, while a new method for detecting spindles is clearly presented, there do not appear to be any new scientific findings in this paper. The authors compare the CNN with only the AT methods for the main result of the paper (Figure 3, Supplementary Figure 3), and not the SNR method. Why not compare directly to the SNR also to see if CNN is actually better? The SNR method is used to generate the data set that the CNN is trained on, as such is it possible for the CNN to do better? Would the AT method be able to pick up more multi-area spindles with a lower threshold?

The CNN and SNR methods are directly compared only for the low vs high memory load task in Supplementary Figure 4. A visual comparison of the low vs high memory load results from the CNN (Figure 3c) and SNR methods (Supplementary Figure 4 top) suggests that the SNR method is equally able to distinguish these conditions. Overall, the advantage of the CNN is not clear.

2. There are several high-level strong claims in the paper that are not directly investigated or supported by the evidence in this paper. For example, "These results thus provide specific neural mechanisms by which memories can be stored in distributed neocortical networks during sleep". "Taken together, these results provide substantial evidence of a specific role for spindles in linking neuron groups distributed widely across cortex during memory consolidation". "The key missing piece is to understand how spindles can guide specific long-range excitatory connections to strengthen during sleep-dependent memory consolidation. We hypothesized that widespread, multi-area spindles might provide this mechansism". At best, the results in this paper provide supportive evidence that spindles could do these things by they do not investigate or establish causality in any way.

3. It is stated that in the ECoG and iEEG data, the AT method detected a subset of spindles that are significantly higher-amplitude than those detected by the CNN methods, using a one-sided Wilcoxon sign rank test. Does this mean that CNN does not detect some of the high amplitude spindles? Is this advantageous? There is something confusing about the way this is stated. A Figure of the distribution in number and amplitude of spindles detected with the 3 methods (CCN, AT, SRN) would be useful.

4. The 3 different recording methods sample activity across different spatial scales, and depth electrodes (iEEG) are sampling vastly different areas (i.e. deeper sources) than ECoG and EEG. As such, it is difficult to relate findings related to "simultaneous spindle detection" in local, regional, multi-area electrodes across these three different measures. A primary concern is that the finding that there are more multi-area spindles (e.g. Figure 2b for iEEG – similar results for EEG and ECoG are not quantified) could be due to volume spread of the spindle source. Is there a way to rule this out? There are ways to minimize the influence of volume conduction with EEG and ECoG source analysis, however, to my knowledge, these methods currently don't exist for iEEG.

Recommendations for the authors:

In Figure 1d, there appear to be other spindle in each of these example traces. Were these not picked up by the algorithm, or simple not highlighted in red? Clarification would be helpful.

Why are the randomly sampled red dashed line in Supplementary Figure 1 not flat? Is the 1Hz filter somehow biasing the amplitude at the center?

It would be helpful to see the results shown in Figure 2b for iEEG for ECoG and EEG data as well.

There is are claims about SNR and CNN being independent of spindle amplitude. Clarification of how the SNR power calculations are independent of amplitude would be useful.

The methods state "We first tested CNN models with different architectures and selected one of the best architectures across sleep recording data sets". What does "one of the best" mean? Quantification would be helpful. Overall, clarification of advantages of this CNN method in identifying rhythmic activity, other than training on rhythmic activity (?), would be helpful.

[Editors’ note: further revisions were suggested prior to acceptance, as described below.]

Thank you for resubmitting your work entitled "Waveform detection by deep learning reveals multi-area spindles that are selectively modulated by memory load" for further consideration by *eLife*. Your revised article has been evaluated by Timothy Behrens (Senior Editor) and a Reviewing Editor.

The manuscript has been improved but there are some remaining issues that need to be addressed, as outlined below:

We thank the authors for carefully considering the feedback they received during the first round of reviews, and for performing additional analyses and modifying their manuscript with the feedback in mind. We also thank the authors for patiently waiting for these reviews. The reviewers found the new version of the manuscript a substantial improvement compared to the first submission. The reviewers feel that additional changes and clarifications can put the authors' contribution in its proper context, and broaden the reach of their contribution in the community. The reviewers think the authors ought to

1. Make their comparisons of the CNN and AT methods in the presence of noise more realistic than currently. Experimentalists would care, for instance, about robustness of the CNN, compared to the AT method, to biological forms of noise (e.g. confounding oscillations such as a brief period of REM theta synched across channels, does the CNN offer an advantage to the AT in that case?). Moreover, the authors seem to have made the comparisons with a fixed noise level/SNR (not to confuse with the 'SNR' method). Does the choice of noise amount have a biological basis? The authors will find text in the reviews unpacking this comment

2. Clarify apparent inconsistencies in certain statistics reported (e.g. spindle rates), as well as conduct statistical tests for some of their data more appropriate than ones currently used. The authors will find text in the reviews unpacking this comment

3. Tone down claims of generality of the CNN, and provide clear explanations for what makes the choice of architecture suitable to the current application to spindle detection. The reviewing editor finds this the most important piece of feedback from the reviews. As written, the reviewing editor feels that the manuscript may promote a culture of using deep learning in neuroscience without understanding how deep learning works. The authors have a unique opportunity of not only presenting an application of deep learning that leads to new scientific findings/hypotheses but also of doing so in a manner that promotes a culture of looking inside the black box and trying to understand it. The authors will find text in the reviews unpacking this comment

*Reviewer #1:*

The manuscript has improved substantially and many of my concerns have been addressed. The deep-learning spindle detection method is more fully reported and the new controls with simulated data (varying noise and amplitude) are an improved approach to compare the CNN and AT methods (Figure S1, new Supplementary Table 1). The new analyses investigating the rotating spindle waves provide a framework for understanding the differences with prior reports of low synchrony.

However, there are several important concerns that should be addressed.

The new comparisons between CNN and AT models using surrogate data with systematically varied noise are useful. However, what was the rationale to compare the specific types of noise presented in Supplementary Table 1? Both the CNN and the AT have similar sensitivities and specificities for all noises, which may suggest that the different noise conditions do not constitute substantially different challenges for the detectors (also suggested by the example in Supplementary Figure 1, in which the spindle has a high signal to noise ratio with all types of noise). I think the readers will be more convinced of the value of the CNN method it can offer an advantage under a 'noise' condition in which the traditional AT methods are likely to struggle. For example, a more interesting source of noise may be the presence of non-spindle oscillations due to brief state changes (e.g., REM or α), or noise artifacts (e.g., examples in Supplementary Figure 12). The sensitivity of traditional methods is likely to go down in examples like these, how does the CNN compare?

Line 92: "co-occurrence"; please clarify the time window or time overlap used to determine co-occurrence. Did any amount of time overlap between spindles count as co-occurrence? Likewise, in Line 150: "simultaneously detected spindles", line 151: "based on co-occurrence". I think it's worth emphasizing in this paragraph that 'co-occurrence' and 'simultaneous' detection do not imply 0-lag synchrony (as discussed in the 'rotating waves' sections).

Lines 102 and 104: "low and high visual memory task" should be "low and high-load…" (also in the Figure 3 legend, and in lines 261-262, 263: 'high-load' visual memory tasks).

There are some inconsistencies in the reported rates (spindles/min) across analyses and figures: Suppl. Figure 4a indicates average spindles/min around 4-5/min in all data types, including EEG data used in the visual memory task. However, the average from the distributions shown in Figure 3b seems much lower, even when combining all spindles across each memory condition, e.g. for the H-VM is about 1/min in local and regional and about 0.5/min for multi-area. In other words, could you clarify how the averages of the distributions in Figure 3b add up to 4 spindles/min in Figure 4a? In addition, in Figure 3b, within the low-VM condition the rate of multi-area spindles is substantially lower (< 0.5/min) than local and regional (about 1/min on average); and within high-VM the rate also appears much lower (<1/min) for multi-area compared to the distributions for local and regional spindles (up to ~2/min). The text indicates p-values for what seems to be a comparison of relative rates (between H-VM and L-VM), which is interesting but is this result due to a significantly lower multi-area spindle rate with low memory? Based on the figure it looks like multi-area spindles occur at lower rates regardless of memory load? It will be helpful to provide statistical comparisons of spindle rates (among all spindle types) within each condition (L-VM, H-VM) so that the relative changes with memory can be interpreted in the context of the absolute spindle rates in each memory condition. Based on the distributions it seems that non-parametric tests would be more appropriate than t-tests.

Suppl. Figure 4: missing (a),(b) labels. Supplementary Figure 4a is not described in the text?

Line 725: 12? sites

Line 154: it'd be useful to reference the Supplementary Table 2 (with the list of the cortical regions).

Lines 170, 171: I believe the verb should be in past tense.

Line 179: The interpretations regarding the association between the visual memory load and subsequent sleep spindles is limited to memory consolidation processes. However, memory performance after sleep was not studied in this work, therefore "the impact on memory consolidation" cannot directly be assessed. Sentences such as these (lines 179-180): "If this is the case, what is the impact of the distribution on sleep-dependent memory consolidation? To answer this question…" could be rephrased to state the key question that the data can address. Other sentences in the manuscript are more accurate and valuable in this respect, e.g., Lines 289-290 "(2) this spatial extent can be modulated by the specific memory conditions prior to sleep (Figure 3)". Indeed, memory consolidation is one of several mechanisms that determine how wakefulness influences sleep oscillations. The authors correctly cover literature on the association between memory and spindles "An increase in spindle density after memory tasks and its relationship with memory consolidation is well established (Clemens et al., 2005; Dang-Vu et al., 2008; Gais et al., 2002; Schabus et al., 2007, 2004)", but the impact of the authors' findings would be enhanced by discussing other hypotheses in the sleep field that are also consistent with the presented results; for example, the high-load working memory condition may increase firing rates and entrain downscaling processes during subsequent sleep (Tononi and Cirelli, 2014; Crunelli et al., 2018; Klinzing et al.,2019).

Paragraph starting 215: is this only with the EEG data?

Figure 4a: missing (a),(b) labels for the plots. Indicate the x,y value of the removed outlier point in parenthesis (in 'a' and 'c') since it's not in the plot but still used in quantifications.

Line 552: "We simulated 60-minutes recording containing" should it be "60-minutes of recording…"?

*Reviewer #2:*

The authors suggest that the lack of algorithms for reliably identifying spindles in neural recordings has led to the under reporting/underestimation of the spatial extent to which spindles occur in the brain. The authors propose an approach based on CNNs that they claim can detect spindles more reliably than existing ones, lead to new insights as to how cross-region spindles may contribute to the integration of 'information' across brain areas.

Strengths

1. The applicability of the proposed methods to neural data from multiple modalities, i.e. EEG, EcOG and iEEG.

2. The framework the authors lay out for constructing high-quality data sets to train the CNNs.

3. The combination of computational methods and new suggested insights into how spindles can facilitate the integration of 'information' across brain areas.

Weaknesses

1. The authors could improve the explanations as to why the CNN seems to do better than legacy methods such as the AT. The explanation surrounding the CNN's ability to detect spindles of different amplitudes does not seem satisfactory enough.

2. The authors claim the applicability of the CNN to data from different modalities, and its generality, as a strength. Given the black-box nature of CNNs, and the lack of an attempt in the manuscript to explain the CNN, its success/failure modes, which aspects of the data it focuses on to detect spindles (e.g. saliency maps), these claims do not feel justified to the reviewer and may contribute to the proliferation of black-box CNNs in neuroscience

Recommendations for the authors:

The reviewing editor found the manuscript a much improved version of the initial submission.

1. The reviewing editor found that the analyses currently in the manuscript do not support the authors' claims of generality of the CNN (e.g. lines 376-377), and that the language surrounding these claims may contribute to supporting an already-widespread tendency to utilize black-box neural networks for analyzing neural data.

a) One feedback from the first round of reviews had to do with the lack of details surrounding the CNN. The reviewing editor appreciates the authors' attempt to improve this. The authors seem to want to emphasize the generality of the architecture. Such claims of generality do not give insight to the reader on how to pick an architecture for detecting oscillatory patterns in a different context (e.g. ripples). The success of the architecture for gravitational wave detection, its use in the current manuscript, do not give license to use it w/o changes in any application.

A practitioner would appreciate guidance on architecture design. Given the length of pattern of interest and a sampling rate, a user would want to know how to pick filter sizes. For instance, for a patter of length 0.5 seconds, a 256 Hz sampling rate, the effective filter (explained below) associated with an architecture ought to have size on order of 100 samples (0.5*256 = 128 samples), not 10 or 1000. The reviewing editor thinks this manuscript could have a much stronger impact if the authors took such questions into considerations. Have authors visualized the filters they learn? Have they considered generating saliency maps to determine which parts of inputs the network focuses on to detect spindles? At present, the emphasis on the generality of the CNN, w/o considerations for what about the authors' specific context makes the choice of architecture suitable (other than the fact that it works for gravitational-way detection) feels a bit disappointing.

More details on guidance on how to design architecture/why current architecture works: For any conv net, the choice of filter size at each layers, together with number of layers relates to size of features one would like to detect. Each layer a conv. Composition of conv equals a conv with a filter size roughly proportional to the sum of filter sizes at each layers. 7x7 filters not uncommon in image proc. Cascading 3-4 such layers (ignoring nonlinearities) gives a filter with an effective 30x30 size.

Concrete suggestions: (a) adding text to the manuscript explaining what what about the authors' specific context makes the choice of architecture suitable (other than the fact that it works for gravitational-way detection), (b) visualizing filters learned by the architecture, (c) consider generating saliency maps.

(b) The reviewing editor feels that the authors can improve the section detailing the training of the CNN. The editor suggests the authors consider using a table to summarize different aspects (number of layers, filter sizes etc….). The editor also suggests that, very early on, the authors mention the subject-dependence of the training.

2. Misc questions

(a) In supplementary figure 4b, the scatter plot for EEG (green) suggests a similar max PSD for the CNN and AT. I would expect a similar rate of spindle detection. Supplementary figure 4a suggests otherwise. Can the authors explain?

(b) In Figure 2c, EcOG the fact that the CNN detects fewer local and regional spindles per minute than the AT method requires explanation. Does this figure refer to correctly detected spindles? Should the editor interpret the lower ratio as the AT method having more false alarms than the CNN?

---

## [Author Response]

[Editors’ note: the authors resubmitted a revised version of the paper for consideration. What follows is the authors’ response to the first round of review.]

Overall, as a group, the reviewers found the application of CNNs to the detection of spindles novel and interesting. More generally, the reviewers thought that a method to detect rhythmic behavior from noisy data could have an important impact on neuroscience research. At the same time, the reviewer expressed some reservations, and felt that the manuscript requires additional work to assess the utility of the CNNs as an addition to sleep biologists' toolkits. Should the authors decide to submit a revised version of the manuscript, we will do our best to have it assessed by the reviewing editor and the reviewers who provided the comments below.

We would like to thank the Editor and both Referees for their thoughtful consideration of the merits and advances in our manuscript. We appreciate that both Referees recognized the novelty and importance of the scientific results; however, they also provided important critiques and comparisons to strengthen this work. We have taken their comments to heart, and in the revised manuscript, we have worked to address them in full.

In summary, we have worked to more fully document the deep-learning waveform detection method we introduce in this work, including more detailed comparisons to previous methods. We have added a new section with simulations using surrogate data and systematically varying noise to demonstrate that our deep learning approach is robust to different types of noise on the recording electrode (new Figure S1). These results indicate that our CNN approach can achieve a very high performance when trained under varying noise and recording conditions (new Table S2). In addition, in our revised manuscript we have gone back to the data from the visual memory task and discovered a novel finding: rotating spindle waves significantly increase following high-load visual memory tasks (new Figure 4 and Video S1). These new results specifically tie the increase in multi-area spindles we reported in the original submission to a neural circuit mechanism for rotating waves in consolidating memories across distributed networks in cortex (see Figure 2, Muller et al., eLife 5, 2016).

We are very grateful to the Editor and Referees for the points raised in the first round of review, and we believe our work to address them has substantially strengthened the manuscript.

The following represent three main critiques/suggestions from the reviews.1. The reviewers found the comparisons to existing methods for spindle detection too simplistic and lacking. More specifically, the authors compare their method mainly to the AT method. The reviewers found the lack of details of the authors' implementation of the AT method, and the lack of thorough comparison to spindle-detection algorithms more sophisticated than the AT method, such as the SNR method, surprising. The reviewers suggest that the authors describe their implementation of the AT method in more detail (e.g. threshold selection etc…). The reviewers also suggest that the authors perform thorough comparisons of the CNN to methods/algorithms more sophisticated than the AT method, both in simulations and on the data used in the manuscript. Evidence that the CNN detects substantially more spindles than these methods would motivate its adoption by biologists.

We appreciate the opportunity to discuss this point. In this work, we focused on the standard implementation of the amplitude-thresholding approach implemented by Mölle et al. (*J Neurosci* 22, 2002). As detailed in a comprehensive analysis by Warby et al. (*Nature Methods* 11, 2014), who performed an exhaustive comparison of six commonly utilized sleep spindle detection algorithms (Bódizs et al., *J Neurosci. Methods* 178, 2009; Ferrarelli et al., *AM. J Psychiatry* 164, 2007; Mölle et al., *J Neurosci* 22, 2002; Martin et al., *Neurobiol. Aging* 34, 2013; Wamsley et al., *Biol. Psychiatry* 71, 2012; Wendt et al., *IEEE Eng. Med. Biol. Soc.* 2012, 2012), these approaches are methodologically similar, relying on bandpass filtering (or wavelets) followed by a thresholding procedure. In this work, we aimed to focus on the signal processing principle common to these approaches: across choices of specific frequency filtering and thresholding procedures, problems can arise from (1) intrusion from signals outside the frequency band of interest (due to bandpass filtering, cf. Figure S12) and (2) missing clearly formed spindle waveforms that fall under the defined threshold. For this reason, we believe that our manuscript presents not only the most pertinent comparison between our CNN approach and the AT method, but also identifies the underlying reason for the new results uncovered by our algorithm. With this in mind, we have clarified in our revised manuscript the general signal processing framework underlying this work: when identifying spindles based on their amplitude characteristics, rather than on waveform characteristics (as with our deep learning approach), the limitations on detecting events in high-noise situations will be general across methods.

We also appreciate the opportunity to clarify our two-step approach for training the CNN algorithm to detect spindles. Importantly, we introduced the SNR algorithm in a previous manuscript (Muller et al., eLife **5**, 2016) as a conservative approach to detect spindles while maintaining a low false-positive rate. This method, which is based on the constant false-alarm rate (CFAR) method in radar (Levanon, N., Wiley, New York, 1988; Minkler, G. and Minkler, J., Magellan, 1990), detects many “true” spindles while minimizing false detections; however, its aim is not to detect all spindles. For this reason, while the SNR approach is excellent for generating a high-quality training dataset and for providing a second check on results from the CNN model, we know that it cannot provide a comprehensive detection method approaching all of the spindles in a recording. In the initial submission, we thus introduced the CNN approach, which detects many more well-formed and high-quality spindles than the AT approach (Figures 2, S3, and S10) and used the CNN approach to reveal multi-area spindles that are selectively modulated by memory load (Figure 3).

We apologize that the implementation of the AT was not clear in the original manuscript. In the revised version, we provide details of implementation including steps taken to compute amplitude using signal envelope over a sliding window of time, setting the threshold based on the existing work in the literature (approximately 2 spindles per minute), and specifying spindle start and end time from the amplitude crossings. We also note that the CNN and AT approach were implemented in as similar a manner as possible to facilitate comparison.

Our revised manuscript provides specific evidence that the CNN detects substantially more spindles than AT. On a set of surrogate data generated to test the robustness of the CNN to different types of noise, the CNN model was able to detect at least 98% of spindle activities, with a specificity above 99%. In contrast, even though the surrogate data does not have any artifact such as line noise, electrical noise, and movement artifacts to introduce signal distortion, the AT achieved lower performance (specificity around 84%-87%). The difference between these methods is even more pronounced in the actual recording, where our comparison reveals that multi-area spindles systematically occur 1.5 (ECoG) to 10 (iEEG) times more often with the CNN than with the AT. Taking these results together, we believe we have provided substantial evidence that the CNN model we have introduced can accurately detect spindles across a range of sleep recording types and provide better performance than the commonly used AT technique.

We appreciate the Referees bringing up these three important points, which we have addressed in our revised manuscript: (1) explaining more clearly how our method compares to other approaches for sleep spindle detection, such as the SNR approach (lines 110-117), (2) clarifying our implementation of the AT approach (Methods – Comparison with amplitude threshold approach), and (3) providing specific evidence that the CNN detects spindles more effectively than the AT approach using surrogate data (Lines 82-86 in the main text, new supplementary section “Methods – Simulated data control – varying noise” and new Figure S1 and Table S1).

We hope these additions fully address the Referees’ concerns.

2. The reviewers felt that the analyses do not support the authors' interpretations and their claims of new scientific findings. First, differences in spatial sampling of neural activity by EEG, EcOG and iEEG, together with the fact that volume conduction may contribute to synchronous activity, limits the interpretation of the authors result. Second, a line of work by Dehghani, Cash, Halgren and colleagues investigating the spatial extent of spindles suggest a lack of synchrony: e.g. (1) doi: 10.1152/jn.00198.2010, (2) doi: 10.1002/hbm.21183, (3) doi: 10.1016/j.clinph.2010.06.018. The reviewers suggest that the authors temper their claims significantly, and put them in the context of the above literature.

We thank the Referees for bringing up these important points, which we have considered very seriously in the process of revision. In our revised manuscript, we have added a new analysis of the spatiotemporal structure in the sleep spindles we detect. In addition to providing novel insight into the lack of synchrony in sleep spindles, this analysis addresses the technical questions raised in this point.

To explain in more detail, we have now added an analysis of spatiotemporal dynamics in the sleep spindles, extending methods developed in our previous work (Muller et al., eLife 5, 2016). For this analysis, we focused on the EEG dataset, with the aim of comparing spatiotemporal dynamics following low- and high-load visual memory tasks. While EEG certainly has reduced spatial resolution compared with the intracranial ECoG and iEEG also studied here, we find that we can robustly detect rotating waves recorded at the scalp (new Figure 4 and Video S1). These waves travel within approximately the same speed as detected in our previous work, which is consistent with the axonal conduction speed of long-range fibers in cortex. Importantly, the observation of clear rotating waves in the scalp EEG directly addresses concerns about volume conduction in the multi-area spindles observed here: the spindles we analyze exhibit tens of milliseconds of offset during individual cycles, and this offset is inconsistent with the effect of volume conduction, which must necessarily be instantaneous. For this reason, it would be difficult for volume conduction effects to be a major confound for our present results on multi-area spindles.

These points open up to a broader and important discussion on synchrony in spindles. We have previously collaborated closely with Dehghani, Cash, and Halgren, and we appreciate the opportunity to discuss the results from analysis of MEG and EEG, which found that correlation measures across sites during spindles was low. Recent work from Birgit Frauscher’s lab studying iEEG depth electrodes has also found similar results (Frauscher et al., *NeuroImage* 105, 2015), notably showing that sites across cortex exhibit broad increases in 11-15 Hz power during spindles detected at the scalp, but exhibit low correlation across electrode sites. In fact, because traveling waves introduce systematic phase offsets across sites, our work can provide an underlying mechanism for these results, accounting for the broad increases of spindle power across cortex (due to multi-area spindles) with low measures of synchrony (due to phase offsets introduced by the traveling wave). Further, with this in mind, it is critical to distinguish measures of synchrony (as in the work of Dehghani, Cash, and Halgren) from the extent of spindle occurrence (as addressed in our work). We have added citations and additional discussion in the main text to highlight and address these points (lines 273-284).

Going further, we can ask about the functional role of these waves. In our previous work, we introduced a specific circuit mechanism for how rotating waves during spindles could facilitate building connections between distributed networks in cortex (cf. Figure 2 in Muller et al., eLife 5, 2016). This mechanism addresses an important point about the connection between sleep spindles and circuit plasticity: if cortical spindles were perfectly synchronized, spikes emitted during one cycle of the spindle oscillation would arrive at their post-synaptic targets with the time delay due to axonal conduction (tens of milliseconds in cortex) (Author response image 1, left), leading always to a pairing within the window for long-term depression (LTD) that would progressively weaken long-range connections (Author response image 1, right). If, however, spindles are self-organized into large-scale wave-like activity patterns, with phase speeds matching those of the underlying fiber networks and stereotyped, precisely repeating trajectories (Author response image 1, left), then excitatory postsynaptic potentials (EPSPs) caused by spikes traveling along pyramidal axons to distant regions in the cortex could align with the local burst of population activity (Author response image 1, right), creating the conditions necessary for both synaptic strengthening and weakening to occur.

**Author response image 1. sa2fig1:** Schematic of spindles and axonal delays. (A) Spikes emitted from region A will arrive at B with a temporal delay of 20 milliseconds (left). If spindle oscillations were perfectly synchronized across the cortex, EPSPs from region A would occur after the spikes in region B, within the window for long-term depression (right). (B) In contrast, if spindles are spatiotemporally organized with stereotyped trajectories (left), then EPSPs from region A would align with population spiking in region B, allowing for synaptic strengthening to occur. Adapted from Muller et al., *eLife* 5, 2016.

This mechanism for building distributed circuits in cortex opens the possibility that pairings between specific sites during spindles, mediated by spatiotemporal organization into traveling waves, can be modulated by memory load. In our revised manuscript, we have compared rotating waves following low and high visual memory tasks in the EEG dataset. Surprisingly, we find that rotating waves in the *temporal -> frontal -> parietal (TFP)* direction are significantly increased during high visual memory tasks (new Figure 4). This result, that rotating waves dynamically change under different memory conditions, provides an additional corroboration that our results are independent of confounds from volume conduction, as the only difference between the two EEG recordings from the same individual and recording setup was memory load. Further, these new results provide intriguing evidence that spatiotemporal dynamics in spindles may adapt to specific memory content during sleep-dependent consolidation. In this case, the significant increase in TFP waves may implicate a set of top-down circuits implicated by the specific memory task; in upcoming work, we aim to understand more fully how the spatiotemporal structure of sleep rhythms changes with memory content during sleep.

We hope that these additional details address concerns about the connection between the observations on sleep spindles in the present manuscript and mechanisms of human memory consolidation. We have added additional details and discussion on this point in the main text (lines 62-71 and 215-271).

Finally, while we consider several very different recording modalities in this work, it is important to emphasize that we do not draw comparisons across different electrode types. For example, we do not compare the rate of multi-area spindles in ECoG with those in iEEG. Rather, we focus on the result which clearly comes out consistently from our analysis: AT-type techniques systematically underestimate the number of multi-area spindles across all types of recordings. The main result of our paper, which is that the CNN model reveals multi-area spindles occur much more often than previously thought, is thus not affected by differences in spatial sampling of the electrodes (Figure 2c and S6b). We apologize that the explanation was not clear in the original manuscript and we have added additional clarification on this point (lines 95-102 and 296-300). We hope these additional clarifications fully address the Referees’ concerns on this point.

3. The reviewers thought the manuscript could benefit from an expanded discussion of the motivation for utilizing the CNN for spindle detection, in particular of how, if at all, they tailored the architecture to the spindle-detection setting. The description of the architecture (layers etc…) provided in the manuscript seems to mirror the description of that developed for gravitational wave detection. More generally, the reviewers thought that a discussion of how to tailor the architecture to the detection of brain rhythms other than spindles would benefit the neuroscience community. For instance, variables such as the length of the rhythmic pattern of interest ought to impact architectural choices.

We thank the Editor and Referees for bringing up this important point. Our work indeed is motivated by the utilization of CNN models in high-noise settings for waveform detection such as earthquakes (Perol et al., 2018) and gravitational waves (George and Huerta, 2018). A range of CNN architectures was proposed in these studies, with 1D to 2D channels as input layers, and convolutional layers ranging from 3 to 8 convolutional layers. We implement an architecture similar to the one proposed by George and Huerta (2018) with small modifications to the input and convolutional layers to take into account the basic features of the spindle rhythm in cortex (e.g. average duration). The CNN architecture is also slightly tailored to different sampling rates in each recording modality. As in previous work, the convolutional layer is designed to start by extracting local features, gradually extracting longer-timescale features by decreasing the feature space. Using this strategy, the CNN model can efficiently learn to detect the specific waveform characteristics of the sleep spindle in different types of recordings. In addition, we verified that the proposed CNN model is not sensitive to the slight change in the number of layers (e.g. 4, 5, and 6 convolutional layers) and hyperparameters such as learning rate, maximum number of epochs used for training, and pooling parameters. Details of these analyses are added in lines 445-466 and 475-488 in the revised manuscript.

We do agree that our two-step combination of the SNR algorithm (which generates a high-quality training dataset) and CNN model (which detects a more complete set of clear, well-formed spindles ranging from large- to small-amplitude events) can be applied more generally to detect brain rhythms other than spindles. We appreciate the Editor and Referees highlighting this potential contribution of our work, and we have added this point to the concluding paragraphs of the revised manuscript (lines 292-300).

Reviewer #1:Mofrad et al. trained a convolutional neural network (CNN) to improve the detection of multi-area spindles and study their regulation by working memory load. They used the CNN algorithm on three different sleep datasets (scalp EEG from humans that had performed a working memory task, intracranial human EEG, and ECoG recorded from macaques). They find that their CNN method identifies more regional and multi-area sleep spindles compared to an amplitude-based spindle detection method. Importantly, they found that the rate of occurrence of regional and multi-site spindles was higher in the sleep after human subjects performed a high-load visual working memory task, compared to after the subjects performed a low-load visual working memory task.The results from this study advance the methods used to detect sleep events. As discussed by the authors, most algorithms to detect sleep oscillations in EEG and LFP data are based on amplitude thresholds; this approach is inadequate when studying regional and global spindles, in which the amplitude may not always remain above threshold in all recording sites. The authors overcome this limitation with a ML approach that detects significantly higher numbers of regional and multi-site spindles. The results presented here demonstrate how ML detection algorithms can offer novel insights on the function of spindles; similar approaches could be helpful with other sleep oscillations. Specifically, the results presented by Mofrad et al. suggest that, compared to amplitude-based detection methods, CNNs improve the detection of spindles that occur temporally correlated across cortical regions. In addition, the authors used the CNN spindle detection method on human sleep data after subjects performed a working memory task. The results are consistent with the hypothesis that cross-regional spindle synchronization is regulated by working memory load prior to sleep, providing insight into how specific cognitive processes influence sleep spindles.An additional strength of this study is the demonstration that the CNN method yields similar results in ECoG, EEG, and iEEG, and in data from macaques and from humans, suggesting that the detection method is applicable across species and across electrophysiological recording methods.A few points should be considered when interpreting the results. The definition of local, regional, and multi-site spindles used in this manuscript is based on the absolute number of electrode sites in which a spindle was detected. Since the ECoG, EEG, and iEEG have different spatial sampling, the definition could overestimate regional and multi-site spindles in iEEG recordings, particularly in the subjects that had large number of electrode contacts in one brain region. This could contribute to the much higher ratio of global spindles observed in the iEEG data compared to ECoG and EEG. Nonetheless, the analyses that support a correlation between widespread spindles and memory load were based on 64-channel scalp EEG and are not expected to be biased by differences in electrode spatial sampling.Lastly, the results are interpreted in the context of the systems memory consolidation hypothesis, which proposes that new memories become integrated into neocortical networks in a process that is facilitated by sleep oscillations. The results from the sleep EEG analyses after subjects performed a low- or high- load visual working memory task are indeed consistent with the idea that multi-site spindles contribute to the consolidation of memories formed before the subjects went to sleep. Likewise, the results are also consistent with the synaptic homeostasis hypothesis, which states that sleep oscillations are important in scaling synaptic plasticity of networks that may have been particularly active during wakefulness. The results of this study are therefore congruent with the possibility of multi-site spindles coordinating the homeostatic regulation of synaptic strength across multiple cortical regions in relation to the cognitive load experienced by the subjects during wakefulness prior to sleep.

We thank the Reviewer for the thoughtful and thorough review, and for highlighting the novelty and importance of the scientific results presented in this work. In our revised manuscript, we have included additional details in order to more clearly document the methods and results presented here. In particular, we have addressed concerns about the difference in spatial sampling across the recordings, added new analysis of the spatiotemporal structure in the sleep spindles we detect, and provided a clear explanation of the connection between our observations on sleep spindles in the present manuscript and mechanisms of human memory consolidation. We hope this additional information and analysis have expanded and substantially strengthened the manuscript.

Recommendations for the authorsThe work presented in this manuscript is highly relevant to advance the methods used to detect sleep oscillations, and to understand the functional significance of sleep spindle coordination across cortical areas. The manuscript text and figures are clear and well-organized. Below are suggestions that may help further improve the quality of this work.

We thank the Reviewer for the positive comments on our submission, including highlighting the importance of developing an accurate, automatic sleep spindle detection algorithm. We also appreciate their helpful comments to improve the paper. In this revised manuscript, we have introduced updates to address all comments in full.

The distinction of three spindle types (local, regional, multi-area) based on the co-occurrence across recording sites should be discussed with regard to the different spatial sampling of EEG, ECoG and, specially, iEEG electrodes. For example, spindles detected in multiple recording sites in subjects such as B and C which have multiple contacts in one brain region (Suppl. Table 1), might be incorrectly labelled as regional or multi-area. Although sampling biases are not expected to affect the results regarding memory, it will be helpful for readers to explain how these potential biases were addressed in the analyses of iEEG data. Could the much higher ratio of multi-area spindles in iEEG data (Figure 2c, right panel) be explained by electrode sampling bias? Along the same lines, it may be more appropriate to refer to 'multi-area' spindles as 'multi-site' spindles if they do not necessarily imply occurrence in distinct cortical areas.

We appreciate the opportunity to discuss this important point. While we consider several very different recording modalities in this work, it is important to emphasize that we do not draw comparisons across different electrode types. For example, we do not compare the rate of multi-area spindles in ECoG with those in iEEG. Rather, we focus on the result which clearly comes out consistently from our analysis: AT-type techniques systematically underestimate the number of multi-area spindles across all types of recordings. The main result of our paper, which is that the CNN model reveals multi-area spindles occur much more often than previously thought, is thus not affected by differences in spatial sampling of the electrodes (Figure 2c and S6b). We apologize that the explanation was not clear in the original manuscript and we have added additional clarification on this point on lines 95-102 and 296-300 in the main text.

In our original submission, we took the multi-area point very seriously, and by identifying the cortical regions for each iEEG contact, we quantified the number of unique cortical regions participating in each spindle (Figure 2b). This analysis demonstrated that spindles classified as “multi-site” by our CNN algorithm do in fact occur across multiple cortical areas (60% of recorded regions, bar plot, Figure 2b), corroborating the “multi-area” designation in our analysis. In our revised manuscript, we have realized the importance of this critical point, and we have introduced updates to highlight it more clearly in the main text (lines 153-159 and 320-321).

In addition, we have expanded on this analysis with new supplementary figures to further support this point. Our new Figure S6a demonstrates that the increase in the percentage of parcellated cortical regions participating in a spindle is significant at the individual level in all five iEEG subjects analyzed in this work. Our new Figure S7 shows these increases remain significant if we consider spindle participation at the level of cortical lobes (frontal, temporal, parietal, occipital) and at the level of cortical systems (executive, limbic, visual, auditory, somatosensory). Both of these analyses provide further evidence to support that spindles detected by the CNN model are distributed widely across multiple brain areas, lobes, and systems, providing insight into the role spindles play in memory consolidation. Finally, at all points results from the CNN model are compared with the AT approach, which tends to detect only local spindles, consistent with our proposition that thresholding-type algorithms may over-emphasize local events (lines 170-173).

It would also be helpful to expand the justification for the choice of CNN algorithms; the text refers to their previous use detecting earthquakes or gravitational waves but is not immediately clear why noise in these applications would be similar to noise in sleep signals.

We thank the reviewer for raising this point, which we have addressed in the main text. Previous works on detecting earthquakes and gravitational waves have demonstrated the potential and efficacy of CNN models for waveform detection in very high-noise settings. Importantly, these methods are relatively general to the type of noise in each recording, provided there is enough training data and high-quality marked events. In this case, the CNN can efficiently learn to detect the specific waveform characteristics distinguishing the sleep spindle rhythm in these recordings. We have added additional clarification on this point on lines 82-86 in the main text.

In addition, to add further support to this point, we have added a new supplementary figure and table in the revised manuscript. By training the CNN model on surrogate data with systematically varying noise properties, we demonstrate that this approach is robust to different types of noise on the recording electrode (Methods – Simulated data control – varying noise and new Table S1 and Figure S1), showing that we can expect this approach to generalize well across recordings with different types of noise. This analysis further supports our validations in the original submission, where we showed that detected spindles exhibit a clear power increase in the expected range (Figure 1b), detected spindles are well-formed in individual examples (Figure 1d), and averaging across detected spindles without bandpass filtering still results in the expected waveform (Figure S3). These additional analyses provide further evidence that the AT has lower detection power even in the surrogate data with no artifact such as line noise, electrical noise, and movement artifacts that can introduce signal distortion in experimentally recorded signals.

Taking these results together, we believe we have provided substantial evidence that the CNN model we have introduced can accurately detect spindles across a range of sleep recording types. We have included text to more clearly highlight the utility and potential applications of this resource in future work (lines 296-300).

Line 35: remove 'stage 2': the references provided are mostly from rodents, in which separate NREM sleep stages are less defined; and spindles are not necessarily exclusive of N2 even in humans.

Thank you for raising this point. We have removed stage 2 (line 39) in the revised manuscript.

Lines 61,62: Long-range connections are one potential link to temporarily coordinate spindles across cortical areas. Another important set of connections that could coordinate spindles are the thalamocortical projections, mainly those thalamic pathways with widespread projections to superficial layers of cortex (references such as Clasca et al., 2012, Eur J Neurosci 35(10):1524-32. doi: 10.1111/j.1460-9568.2012.08033.x.).

We thank the Referee for bringing this important point to our attention. We have added the thalamocortical connections as a potential pathway for coordination of spindles in the revised manuscript (lines 246-254).

Lines 102-103: please specify how descriptive statistics are reported; are the values reported in parenthesis the mean and standard deviations? Clarify in the methods section how duration was calculated.

We thank the Reviewer for this and apologize that it was not clear in the original submission: the descriptive statistics here are average ± SEM (line 124). We have added detailed explanations to clarify the calculation of spindle duration in the supplementary section “Methods Convolutional neural network (CNN) for sleep spindle detection” (lines 480-488). We have also clarified the calculation of spindle duration, which is included in our response to the next comment.

The methods state that 'The CNN model takes a window of sleep recording (500 ms which is bandpass…)' (Line 303), is that a 500ms moving window? With what step size? What determines the start and end of a spindle in the CNN and AT methods?

Thank you for bringing up this important point. We have added a detailed explanation of the entire process in the supplementary section “Methods – Convolutional neural network (CNN) for sleep spindle detection” (lines 480-488) and “Methods – Comparison with amplitude-threshold approach” (lines 531-537).

Briefly, the CNN model takes a sliding window of sleep recording (500 ms, signal bandpassed 1-100 Hz) as an input and determines its label (spindle or non-spindle). The sliding window starts at the beginning of the recording and moves 100 ms in each step. To find the start and end time of a spindle, we first combined all overlapping spindle windows. We then included neighbor windows if there is any spindle within 100 ms of the combined windows to account for potentially mislabeled windows. The start of a spindle is finally set to the beginning of the first window, and the end of the spindle is set to the end of the last window.

To create a clear comparison, we implement precisely the same procedure for the AT approach. To do this, we compute the signal envelope over a sliding window of 500 ms with 100 ms step size. We then compare signal envelope for each window to the AT threshold. Finally, we combine windows as with the CNN approach to get the start and end time of each spindle.

Line 161: include p value for local spindles (as done in the figure legend).

We appreciate this suggestion, and we have added the local spindle p-value to the main text

(lines 198-199).

Given the unique dataset and methods, and the focus of this work on the locality and globality of spindles, the authors may want to report on potential changes in the density of local spindles specifically over visual and frontal cortical areas. Is it possible that increases in local spindle density are not significant when all regions are considered together but may be significant in certain regions engaged by the visual working memory task? In that sense, it is interesting that the SNR approach detects a (smaller) increase in local spindles with high memory load (Supp. Figure 4a). A more precise analysis of local spindles over relevant cortical areas will not diminish the author's main result on multi-area spindles, but it will provide additional cues on the role of spindle spatial synchronization in sleep and will enhance the value of this interesting work.

We thank the Referee for suggesting this new analysis which provides additional insight on the role of spatial synchronization of spindles during sleep. Following this suggestion, we studied the change in density of spindles in frontal, occipital and parietal lobes (new Figure S9) in low and high visual memory conditions. To do this, in each cortical lobe, we computed the percentage of electrode sites with spindles within the detected windows by the CNN model. Interestingly, we observe a significant increase in the percentage of electrode sites in high versus low visual memory conditions across all cortical lobes. On average the occipital lobe with more than 20% electrode-site participation has the highest and the frontal lobe with about 10% has the lowest participation (lines 207-213).

Lines 241-242 say that 'the quality of sleep was studied by the degree of spatial synchronization'; it will be helpful to clarify this statement because if the degree of spatial synchronization was used to select the sleep data used for this work, the dataset may include more synchronized activity compared to other datasets. This does not nullify the results, but clarification will be important to ensure replicability. Likewise, the methods used for sleep detection should be described in more detail, was sleep detected in the same way in all datasets?Line 250: only visual inspection was used to detect stage 2 in the EEG data? clarify if only NREM sleep was used in all datasets used for analysis? (same for the iEEG recordings)

We thank the Referee for noting this important point on selecting sleep epochs in the recordings. In our analysis, we studied the full recordings annotated as sleep without excluding REM states. As mentioned in the Methods section, sleep state was assessed in the recordings using a method best adapted to each recording type. For example, in the EEG recordings, sleep state was manually annotated by an expert for stage 2 NREM sleep. In the iEEG recordings, sleep annotations were provided by hospital staff who observed times when a patient was sleeping and noted in their chart. Finally, in the ECoG dataset, NHP subjects were placed in a quiet and dark room with eyes covered for up to 90 minutes. The sleep state in this case was determined using the degree of spatial synchronization and power increase in the 1-4 Hz δ frequency band.

We appreciate the Reviewer’s comment on this point, and in developing this analysis, we carefully considered the differences in determining sleep state across the datasets analyzed here. Because in the ECoG data spatial synchronization was only assessed in the 1-4 Hz frequency band, which is known to be consistently more synchronized during sleep (Destexhe et al., *J Neurosci* 19, 1999; Dang-Vu et al., *Neuroimage* 28 2005; Chauvette et al., *J Neurosci* 31 2011; Murphy et al., *PNAS* 106.5 2009), we believe that these datasets are not more synchronized than other sleep recordings. Further, we find consistent results across the datasets we analyzed, which varied in the way they detected the sleep state. With this in mind, we believe our analysis provides considerable evidence to indicate that this potential concern does not affect our results.

We fully agree that more details would be helpful to document the recordings, and we appreciate the Reviewer’s suggestion along these lines. In our revised manuscript, we have included several updates to the supplementary section “Methods – Recordings” to detail the behavioral and electrophysiological aspects of these datasets (lines 372-373, 383-386 and 393-394). We hope this additional information also addresses the Reviewer’s more general point about reproducibility, which is an important commitment in our work.

Not clear what is added by the paragraph between Lines 283-288 unless more details are provided. If the model simulates spiking activity in the awake state, is this helpful as a ground truth control for sleep LFP? What type of spiking model? What network architecture?

We thank the Reviewer for noting that more details are necessary to clarify the spiking network control simulation. With this analysis, we sought to provide additional evidence that our approach can robustly detect sleep rhythms while also – importantly – providing no detections when sleep-like rhythms are not present. To do this, we used data produced by a spiking neural network model we have currently developed to study the spontaneous background activity of awake cortical circuits (Davis, Benigno, et al., *Nature Communications*, 2021 [link]). This model, composed of several million neurons with biologically realistic synaptic connectivity (several thousand synapses per cell), creates realistic ongoing activity patterns consistent with the well-studied asynchronous-irregular state. In addition, we have incorporated a recently developed LFP proxy (Mazzoni et al., *PLoS Comput Biol*, 2015) in the model, allowing us to analyze a population signal using our spindle detection algorithm. Starting with the first step in this process, we first apply the SNR algorithm to detect well-formed spindles for training data. As expected, however, the SNR approach does not detect any spindles in the simulated data.

Because the simulation lacks the mechanisms for generating large-scale thalamocortical and corticocortical rhythms, this provides a ground-truth test that our approach does not detect spurious spindles in neural population signals. The results from this analysis provide additional confirmation, in a setting where we know rhythms such as spindles do not exist, that our approach is not only robust to various types of noise but also sensitive to spindle rhythms embedded in population signals. We have added additional details about this model in the supplementary section “Methods – Signal-to-noise ratio (SNR) measure for sleep spindle detection” (lines 430-435).

Paragraph starting in line 290. Was the CNN model trained and optimized for each dataset?Lines 296-297: 'best' based on what?

This is an excellent point, and we thank the Reviewer for bringing this up. In the revised manuscript, we have included additional details on the process of training and optimization of the CNN model. We implement an architecture similar to the one proposed by George and Huerta (2018) with small modifications to the input and convolutional layers to take into account the basic features of the spindle rhythm in cortex (e.g. average duration). The CNN architecture is also slightly tailored to different sampling rates in each recording modality. As in previous work, the convolutional layer is designed to start by extracting local features, gradually extracting longer-timescale features by decreasing the feature space and increasing channels. Using this strategy, the CNN model can efficiently learn to detect the specific waveform characteristics distinguishing the sleep spindle rhythm in different types of recordings. In addition, we verified that the proposed CNN model is not sensitive to the slight change in the number of layers (e.g. 4, 5, and 6 convolutional layers) and hyperparameters such as learning rate, maximum number of epochs used for training, and pooling parameters using similarity measure across the predicted values. To perform this sensitivity analysis, for each CNN architecture, we made a grid search over the potential range of hyperparameters, measuring the similarity of model output by Cronbach's α. Similarity across hyperparameters within 10-50% of those used in our analysis was greater than 0.99, indicating high reproducibility under moderate parameter variability. We then used the same architecture and hyperparameters across all subjects and recording datasets. We trained a separate CNN model for each subject on a portion of the available recording and then applied the trained model to detect spindles across the entire recording.

We thank the Reviewer for pointing out the importance of including more details on this point. We have added additional material in lines 445-466 and 475-488 of the revised manuscript to clarify the optimization, parameter tuning, training, and implementation of the CNN.

Line 350 states that the overall threshold for the AT method was based on the average root mean squared. Was the AT approach developed and applied independently for each data set (EEG, ECoG, iEEG)? It is not clear from the text if the different signal amplitudes in the datasets were considered for AT detection (were the data and RMS normalized?).

This is an important point for the clarity of the manuscript, and we appreciate the Reviewer for bringing this up. As with the CNN model, we applied the AT algorithm independently for each subject to account for the difference across subjects and recordings. We have updated the Methods on line 541-543 to clarify this point.

Figure Supp. 5: correct typo in the figure legends (spindle)

Thanks for catching this typo. We have corrected the error in the revised manuscript (Figure S10).

Reviewer #2:The primary goal of this paper is to provide evidence that sleep rhythms are more spatially extended than previously known, using a deep learning convolutional neural network (CNN) model specifically designed to characterized rhythmic activity that has not previously been applied to sleep data.Strengths1. The authors establish that the CNN method proposed can detect spatially extended sleep rhythms and that difference occur when measuring spindles after a low vs high load memory tasks in multiple data sets.2. The CNN method presented may provide a powerful technique to characterize a range of brain rhythms and difference across tasks or patient populations.3. The results and figures are clearly presented.4. The methods are sound, with the exception that more detail on the method and how it uniquely accounts for rhythmic activity is warranted and should be discussed in the Results section of the paper, rather than in the methods.

We thank the Reviewer for these positive comments on the methods and results presented in this work. In our revised manuscript, we have included additional detail in order to more clearly document the methods in this work. In particular, we have added more specific characterization of how the CNN model specifically and accurately detects sleep spindle activity in the main text and supplement (lines 82-86, and 95-102 and, Figure S1 and Table S1).

Weaknesses1. The results of the paper do not establish that this CNN method is better than prior methods at detecting spatially extend sleep rhythms, or that the methods is better able to distinguish sleep spindles after a low vs high load memory task. As such, while a new method for detecting spindles is clearly presented, there do not appear to be any new scientific findings in this paper.

We appreciate the Referee's point, which we considered carefully in preparing our revised submission. The main focus of our work is to compare our CNN approach to standard amplitude-thresholding approaches. The hypothesis underlying this comparison is that amplitude-based approaches, in general, will miss many spindles that exhibit transient dips in power, or those that are very clearly formed but small in amplitude (see Figure S5). While the simplest implementation of this algorithm (Mölle et al., *J Neurosci* 22, 2002) can be improved by imposing additional conditions on the AT procedure (e.g. Hagler et al., *J Neurosci* 38, 2018 and Warby et al., *Nature Methods* 11, 2014), it remains difficult to cover all cases by successively adding additional conditions. Training a CNN model to detect spindles, however, has the advantage of developing these conditions automatically, without requiring hand-tuning by the analyst.

We thank the reviewer for bringing up this point about clearly comparing the CNN method to prior approaches, and we have now highlighted this more clearly in the revised manuscript. In addition, in the original submission we directly compared the changes in sleep spindles following high- versus low-load visual memory tasks (Figure 3b and S8b), and we found that the CNN was able to detect a significant increase, while the amplitude-thresholding approach found no change. We believe this result clearly demonstrates the potential for our CNN approach to find new scientific results in analyses of sleep rhythms, and we have now highlighted this more clearly in the revised text.

Finally, in our revised manuscript, we have now gone back to the recordings from the visual memory task and discovered a novel finding about spatiotemporal dynamics during multi-area spindles: rotating spindle waves significantly increase following high-load visual memory tasks, which we had previously observed to be associated with increased multi-area spindles (new Figure 4 and Video S1). These new results specifically tie the increase in multi-area spindles we reported in the original submission to a neural circuit mechanism for rotating waves in consolidating memories across distributed networks in cortex (see Figure 2, Muller et al., eLife 5, 2016). We have highlighted these new results in our responses to the main critiques, and also revised the manuscript to reflect these points (lines 62-71, 104-107 and 215-271 and new Figure 5).

In summary, we hope that these additions clearly and sufficiently address the Referee’s original concerns about our manuscript. We believe that the new computational approach presented in this work is clearly more effective at detecting multi-area spindles, which are not observed using the previous approach, and at finding new scientific results, such as the selective increase in multi-area spindles and rotating waves following challenging memory tasks.

The authors compare the CNN with only the AT methods for the main result of the paper (Figure 3, Supplementary Figure 3), and not the SNR method. Why not compare directly to the SNR also to see if CNN is actually better? The SNR method is used to generate the data set that the CNN is trained on, as such is it possible for the CNN to do better? Would the AT method be able to pick up more multi-area spindles with a lower threshold?

We thank the Reviewer for bringing up this important point. We introduced the SNR algorithm ourselves in a previous manuscript (Muller et al., eLife 5, 2016) as a conservative approach to detect spindles while maintaining a low false-positive rate. This method, which is based on the constant false-alarm rate (CFAR) method in radar (Levanon, N., John Wiley and Sons, New York, 1988; Minkler, G. and Minkler, J., Magellan, Baltimore, 1990), detects many “true” spindles while minimizing false detections; however, its aim is not to detect all spindles. For this reason, while the SNR approach is excellent for generating a high-quality training dataset (as used here for the CNN model, Figure 1) and for providing a second check on results from the CNN model (as in Figure S10), we know that it cannot provide a comprehensive detection method approaching detection of all spindles in a recording. For this reason, we have adapted the deep learning waveform detection approaches to create the method we introduce in this work. The advantage of the CNN approach we introduce is that it can detect many more spindles at high quality.

Unfortunately, because of potential artifacts associated with the AT approach, decreasing the amplitude threshold in that method will not only detect more spindles, but also more mis-classified activity patterns (i.e. false positive detections). For example, we can start with an amplitude threshold that produces 2 spindle detections per minute (Gaïs et al., J Neurosci, 2002) and then reduce this threshold by half. In Author response image 2, we provide several examples of clearly non-spindle activity detected using this reduced threshold.

**Author response image 2. sa2fig2:** Impact of AT threshold on spindle quality. (a) Plotted are the average time-shifted over spindle detected by the AT algorithm using 2 spindles per minute threshold (black line), additional spindles detected by the AT algorithm after decreasing the threshold to its half (blue line), and lastly a subset of randomly matched non-spindle windows (red line). As expected by decreasing the threshold, the average over detected spindle activities starts to drift away from the expected 11-15 Hz oscillatory structure. (b) Examples of non-spindle activity (blue line) detected by the AT approach when we decrease the threshold producing 2 spindles per minute (solid green line) by half (dashed green line). The red line is the signal envelope.

In contrast to reducing the threshold in the AT approach, however, our CNN method is able to detect high-quality spindles that co-occur across the cortex much more often than previously thought. In our original submission, we validate this approach through analyzing the power spectrum of detected events (Figure 1b), through a time-aligned averaging approach (Supplementary Figure S3), and finally through comparison to the subset of spindles detected by the AT approach (Figure S5). At the same time, we appreciate that we may not have made the importance and impact of the CNN and SNR approaches clear in the original submission. In our revised submission, we have added additional text clarifying the SNR algorithm we have introduced previously, its use for generating a high-quality training dataset for the CNN (lines 110-117 and 292-300 in the main text), and its use in validating the results from the CNN. We appreciate that these points may not have been clear in the original submission, and we hope these new additions fully address the Referee’s concerns.

The CNN and SNR methods are directly compared only for the low vs high memory load task in Supplementary Figure 4. A visual comparison of the low vs high memory load results from the CNN (Figure 3c) and SNR methods (Supplementary Figure 4 top) suggests that the SNR method is equally able to distinguish these conditions. Overall, the advantage of the CNN is not clear.

We thank the Referee for raising this important point, which may not have been clear in the original manuscript. As detailed in the previous response, the main disadvantage of the SNR approach is that it is overly conservative: it will exclude too many spindles at the expense of keeping false positives low. Thus, while the SNR approach is useful for quickly identifying a set of spindles with low false-positive rate, it is not adequate for studying the full population of spindles or for answering questions about changes in distribution in general.

At the same time, however, the fact that the subset of spindles detected by the SNR approach exhibits the same increase under high versus low visual memory load as observed by the CNN approach provides an important corroboration of this result. These two approaches provide clearly converging evidence that distributed spindles appear following high visual memory tasks, a change that is not detected by the AT approach. For this reason, we believe that we have not only validated the performance of the CNN approach in contrast to amplitude-based approaches, but also clearly demonstrated that this approach is able to find qualitatively new results providing insight into the process of human memory consolidation.

We again thank the Referee for raising this point, which we have tried to clarify in the main text (lines 202-207). We believe that this additional explanation more clearly introduces the advantages of the CNN approach and addresses the Referee’s point in full.

2. There are several high-level strong claims in the paper that are not directly investigated or supported by the evidence in this paper. For example, "These results thus provide specific neural mechanisms by which memories can be stored in distributed neocortical networks during sleep". "Taken together, these results provide substantial evidence of a specific role for spindles in linking neuron groups distributed widely across cortex during memory consolidation". "The key missing piece is to understand how spindles can guide specific long-range excitatory connections to strengthen during sleep-dependent memory consolidation. We hypothesized that widespread, multi-area spindles might provide this mechansism". At best, the results in this paper provide supportive evidence that spindles could do these things by they do not investigate or establish causality in any way.

We thank the Referee for raising this point, with which we fully agree, and we apologize for any ambiguity in the original submission. This manuscript was originally meant as a follow up to a previous article (Muller et al., *eLife* 5, 2016), in which we reported that spindles were often organized as waves rotating across the cortex. In the previous manuscript (and in additional modeling work), we studied the phase speed distributions of the traveling spindle waves, which fell in the range of 2-5 m/s. As these speeds fall uniquely within the conduction velocity of the long-range fibers in cortex, we proceeded to study the long-range fibers as a potential propagation substrate and introduced a neural circuit mechanism for how these waves can enable the consolidation of memories across sets of neurons distributed widely across cortex.

We appreciate the Referee’s point and again apologize that this important connection was not clear in the original manuscript. We have taken the Referee’s concern to heart, and in the revised manuscript, we have connected our analysis of multi-area spindles to our previous results on spindle traveling waves (new Figure 4). We specifically have gone back to the data from the visual memory task and discovered a novel finding: rotating spindle waves significantly increase following high visual memory tasks, which we had previously observed to be associated with increased multi-area spindles. These new results specifically tie the increase in multi-area spindles we initially reported in the original submission to a neural circuit mechanism for rotating waves in consolidating memories across distributed networks in cortex (see Figure 2, Muller et al., *eLife* 5, 2016). We added new analysis in line 215-271 in the main text and the details of the computation in the new sections, “Methods – Rotating wave direction” and “Methods – Simulated data control – rotating wave”, in the supplement.

3. It is stated that in the ECoG and iEEG data, the AT method detected a subset of spindles that are significantly higher-amplitude than those detected by the CNN methods, using a one-sided Wilcoxon sign rank test. Does this mean that CNN does not detect some of the high amplitude spindles? Is this advantageous? There is something confusing about the way this is stated. A Figure of the distribution in number and amplitude of spindles detected with the 3 methods (CCN, AT, SRN) would be useful.

We appreciate this point, and in the revised submission we have this in the text to make the point more clear. While the AT approach detects only the highest amplitude spindles in the dataset (new Figure S4a), the CNN approach is able to detect some lower amplitude (but still clearly formed) spindles in addition. Thus, as expected, this result is reflected in the fact that the set of spindles detected by the AT approach (which preferentially detects the highest-amplitude events) has higher average amplitude than those detected by the CNN (which detects a broader set of spindles). Specific examples of this can be observed in the EEG recordings in Figure 1c, where the first and second spindles were not detected by AT and in Figure S5, which provides examples where the CNN detects clear spindles but the AT does not.

Moreover, to address the Referee's concern regarding the number of spindles, we perform a new analysis, in which we computed the number of spindles per minute for each session across all recordings. Interestingly, the CNN model tends to detect on average 4-5 spindles per minute (almost twice of what was reported in the literature) as opposed to the AT which resulted in 2-3 spindles per minute except in the NHP ECoG recording (new Figure S4b). The result further demonstrates that if trained properly, the CNN model has the ability to detect hidden spindles that are being unnoticed and provides a great opportunity to study the spatial and temporal analysis of spindle activities across the cortex.

We thank the Referee again for raising this important point, which we have attempted to clarify in the revised submission. We have specifically focused on clarifying why spindles detected with the AT method are expected to have higher amplitude overall (lines 138-147) and how the CNN method, which detects spindles missed by the AT approach, improves our understanding of sleep spindles and changes following visual memory tasks (lines 202-207 and 286-292). We believe these additional points fully address the Referee’s concern on this point.

4. The 3 different recording methods sample activity across different spatial scales, and depth electrodes (iEEG) are sampling vastly different areas (i.e. deeper sources) than ECoG and EEG. As such, it is difficult to relate findings related to "simultaneous spindle detection" in local, regional, multi-area electrodes across these three different measures. A primary concern is that the finding that there are more multi-area spindles (e.g. Figure 2b for iEEG – similar results for EEG and ECoG are not quantified) could be due to volume spread of the spindle source. Is there a way to rule this out? There are ways to minimize the influence of volume conduction with EEG and ECoG source analysis, however, to my knowledge, these methods currently don't exist for iEEG.

We thank the Referee for raising this critical point. We agree that it is important to carefully consider the differences in spatial sampling by the ECoG, EEG, and iEEG electrodes studied in this work. For this specific point, it is important to note that iEEG depth electrodes record cortical activity with the highest spatial resolution possible of intracranial recording techniques (Mukamel and Fried, *Ann Rev Psych* 63, 2012), while also quickly becoming standard in the field of neurosurgery because of their increased safety and improved patient outcomes compared to subdural grid arrays (Tandon et al., *JAMA Neurol* 76, 2019). Because iEEG electrodes have the highest spatial resolution of the recording techniques considered here, with ECoG (Dubey and Ray, *J Neurosci* 39, 2019) and EEG (Babiloni et al., *Clin Neurophysiol* 112, 2001) representing increasingly lower spatial resolution, it would be difficult for the higher number of multi-area spindles observed in iEEG to result from increased volume spread of the spindle source, as the influence of volume conduction is minimized in this recording technique. Rather, if increased volume spread were the underlying cause for this finding, we would expect the pattern to be the opposite, with more multi-area spindles in the EEG data (where volume conduction effects are highest). We thank the Referee for raising this important point, which we have clarified in the revised text (lines 167-169).

Recommendations for the authors:In Figure 1d, there appear to be other spindle in each of these example traces. Were these not picked up by the algorithm, or simple not highlighted in red? Clarification would be helpful.

We thank the Referee for pointing this out and apologize for any confusion. In the original figure, we only highlighted the spindles in the central window (at the middle of the plot). In the revised Figure 1d (line 304), we have now highlighted all detected spindles, in order to demonstrate more clearly the events detected by our CNN approach.

Why are the randomly sampled red dashed line in Supplementary Figure 1 not flat? Is the 1Hz filter somehow biasing the amplitude at the center?

We thank the Referee for bringing this point up. In Figure S3, we verify that spindles detected by the CNN approach exhibit high-quality waveforms and rule out potential artifacts from bandpass filtering. To do this, we took each signal highpassed at 1 Hz, re-aligned to the largest oscillation peak in the time window, and then averaged across detected events. With this procedure, the average across detected spindles exhibits clear 11-15 Hz oscillations, even without bandpass filtering. This calculation thus provides strong evidence that the CNN approach detects high-quality spindle events.

We then repeated this calculation on a matched number of randomly selected non-spindle windows, which are those plotted as red dashed lines in Figure S3. Because we align signals to the central peak in the window, we expect there to be a positive peak at this point. Naturally, this peak will exhibit a decay consistent with the autocorrelation time present in the 1 Hz filtered signal. Importantly, however, this control condition exhibits no 11-15 Hz oscillations (i.e. additional peaks beyond the central one), demonstrating that the CNN method correctly identifies non-spindle windows, as well.

We again thank the Referee for raising this point. In the revised submission, we have included a few sentences clarifying this point in the main text (lines 136-138) and in the supplementary section “Methods – Time-shifted averaging control” (lines 509-514).

It would be helpful to see the results shown in Figure 2b for iEEG for ECoG and EEG data as well.

This is an important point, and we appreciate the Referee bringing this up. For the analysis in Figure 2b, we developed an atlas-based computational approach to placements of iEEG electrodes in collaboration with a neuroimaging laboratory. This approach included white and gray matter segmentation, registration, and cortical segmentation of iEEG electrodes. Unfortunately, we could not perform similar parcellation in EEG and ECoG recordings but we were able to further confirm that multi-area spindles are in fact distributed widely across the brain in the EEG dataset. To do this, we studied cortical lobe participation (new Figure S9) and using electrode coordinates we analyzed the change in rotational pattern of multi-area spindles (new Figure 4) in the EEG.

There is are claims about SNR and CNN being independent of spindle amplitude. Clarification of how the SNR power calculations are independent of amplitude would be useful.

Thank you for bringing up this concern. We introduced the SNR algorithm in a previous manuscript (Muller et al., *eLife* 5, 2016) as a conservative approach to detect spindles while maintaining a low false-positive rate. This method is based on the constant false-alarm rate (CFAR) method in radar (Levanon, N., *Wiley*, New York, 1988; Minkler, G. and Minkler, J., *Magellan*, 1990) which is used for narrow-band rhythmic activity detection in a high-noise situation with the goal of reducing false-positive detection. The CFAR is an adaptive process for target detection which adjusts the detection threshold accordingly (Weinberg, Graham Victor, *IET Radar, Sonar and Navigation* 7.2, 2013) to account for the change in the noise power/amplitude. Inspired by this technique, the SNR algorithm detects spindle activities by measuring the ratio of power within the frequency band of interest (here, 9-18 Hz) to power in the rest of the spectrum (1-100 Hz bandpass, with band-stop at 9-18 Hz) which make this technique approximately amplitude-invariant. While conducting our analysis, we observed a lot of examples which verified our claim of SNR being approximately amplitude invariant. An example is presented in Figure 1d in the main manuscript. In this example, the spindles detected in the EEG recording contain high and low amplitude spindles. All of these spindles were detected by both the CNN model and SNR approach, while the AT approach fails to detect the first (low-amplitude activity) and last spindles (temporary drop in amplitude) (Author response image 3) (Lines 549-550).

**Author response image 3. sa2fig3:** AT sensitivity to amplitude. The AT algorithm fails to detect clearly formed spindles detected by the CNN model in Figure 1d EEG recording because of low amplitude or temporary drop of amplitude below the threshold.

To further validate this claim, we performed this new analysis in which we used surrogate data with generated signals containing spindles and additive noise (of several types, including white and brownian noise). We then utilized the surrogate data to study how sensitive the SNR approach is to the change in spindle and noise amplitude as opposed to the AT techniques (lines 552-575). We first implemented both approaches on the surrogate data and detected spindle activities. We then repeated the entire analysis, once after doubling the noise amplitude and once after dividing the spindle amplitude in half. Lastly, using an example spindle detected by both approaches in the original setting, we showed how robust the SNR algorithm is in face of change in the signal/noise amplitude (new Figure S2). We added the details of the computation in Lines 110-117 and the new supplementary section, “Methods – Simulated data control signal amplitude”.

The methods state "We first tested CNN models with different architectures and selected one of the best architectures across sleep recording data sets". What does "one of the best" mean? Quantification would be helpful. Overall, clarification of advantages of this CNN method in identifying rhythmic activity, other than training on rhythmic activity (?), would be helpful.

We thank the Referee for pointing this out and apologize that the explanation was not clear in the original manuscript. Starting from the model architectures developed in previous work, we implement an architecture similar to the one proposed by (George and Huerta, 2018) with small modifications to the input and convolutional layers to take into account the basic features of the spindle rhythm in cortex (e.g. average duration). The CNN architecture is also slightly tailored to different sampling rates in each recording modality. We verified model quality using ECoG recordings by minimizing the difference between predicted and training labels marked by the SNR approach. In addition, we confirmed that the proposed CNN model is not sensitive to the slight change in the number of layers (e.g. 4, 5, and 6 convolutional layers) and hyperparameters such as learning rate, maximum number of epochs used for training, and pooling parameters using similarity measure across the predicted values. To perform this sensitivity analysis, for each CNN architecture, we made a grid search over the potential range of hyperparameters, measuring the similarity of model output by Cronbach's α. Similarity across hyperparameters within 10-50% of those used in our analysis was greater than 0.99, indicating high reproducibility under moderate parameter variability. We then used the same architecture and hyperparameters across all subjects and recording datasets. We trained a separate CNN model for each subject on a portion of the available recording and then applied the trained model to detect spindles across the entire recording (Lines 445-466).

The main advantage of our CNN method is its ability to identify spindles based on waveform characteristics, rather than their amplitude or power (which causes limitations on detecting low-amplitude events in high-noise situations) (Figure 1 and S4). Furthermore, the CNN method can be generalized well across recordings with different types of noise, provided there is enough training data and high-quality marked events. Previous works on detecting earthquakes and gravitational waves have demonstrated the potential and efficacy of CNN models for waveform detection in very high-noise settings. We further validate this claim by training the CNN model on surrogate data with systematically varying noise properties. We demonstrate that this approach is robust to different types of noise on the recording electrode (new Table S1 and Figure S1). Taking these points together, we believe that our two-step combination of the SNR algorithm (which generates a high-quality training dataset) and CNN model (which detects a more complete set ranging from clearly formed large- to small-amplitude activities) can be applied more generally to detect brain rhythms.

We thank the Reviewer for pointing out the importance of including more details on this point. We have added additional material in lines 82-86 and 292-302 in the main text and the supplementary section, “Methods – Convolutional neural network (CNN) for sleep spindle detection”, and “Methods – Simulated data control – varying noise” of the revised manuscript to clarify the optimization, parameter tuning, training, and implementation of the CNN.

[Editors’ note: what follows is the authors’ response to the second round of review.]

We thank the authors for carefully considering the feedback they received during the first round of reviews, and for performing additional analyses and modifying their manuscript with the feedback in mind. We also thank the authors for patiently waiting for these reviews. The reviewers found the new version of the manuscript a substantial improvement compared to the first submission. The reviewers feel that additional changes and clarifications can put the authors' contribution in its proper context, and broaden the reach of their contribution in the community. The reviewers think the authors ought to

We would like to thank the Editor and the Referee again for their valuable and thoughtful comments, which have substantially strengthened this paper. We appreciate that both the Editor and Reviewers find the revised manuscript to be substantially improved. In this new revision, we have performed additional analyses and provided more details on our approach. We hope the additional material provided further strengthens our manuscript and addresses all remaining critiques and comments in full.

1. Make their comparisons of the CNN and AT methods in the presence of noise more realistic than currently. Experimentalists would care, for instance, about robustness of the CNN, compared to the AT method, to biological forms of noise (e.g. confounding oscillations such as a brief period of REM theta synched across channels, does the CNN offer an advantage to the AT in that case?). Moreover, the authors seem to have made the comparisons with a fixed noise level/SNR (not to confuse with the 'SNR' method). Does the choice of noise amount have a biological basis? The authors will find text in the reviews unpacking this comment

We thank the Editor and Reviewers for raising this important point and suggesting a new analysis to provide additional evidence on the advantages of using the CNN model than the traditional AT approach. The logic behind the noise sensitivity analysis was indeed to demonstrate improved performance of the CNN model under different noise types and magnitudes. Following the Editor and Reviewer's suggestion, we studied the change in the performance of the CNN and AT approaches under biological forms of noise, such as theta oscillations, and non-biological artifacts, such as line noise, electrical noise, and movement artifacts. On the new set of surrogate data containing simulated theta oscillations as confounds, the CNN model was able to detect at least 97% of spindle activities, with a specificity above 99%, while the AT achieved lower performance (specificity around 72%). In particular, the AT approach seems to be very sensitive to recording artifacts. Our new analysis demonstrates the CNN model performs robustly in face of increases in the number of artifacts, while the performance of the AT substantially decreases as the number of artifacts per minute increases(new Figure 1—figure supplement 6). These new analyses further confirm that the CNN approach introduced here can accurately detect spindles across a range of sleep recording types and provide better performance than the commonly used AT technique.

We appreciate the Editor and Reviewers raising this important point, which we have addressed in our revised manuscript by providing additional evidence that the CNN detects spindles more effectively than the AT approach using surrogate data (lines 166-170 in the main text, updated supplementary section “Simulated data control – varying noise” and new Figure 1—figure supplement 6 and updated File S1). We hope these additions fully address the Referees’ concerns.

2. Clarify apparent inconsistencies in certain statistics reported (e.g. spindle rates), as well as conduct statistical tests for some of their data more appropriate than ones currently used. The authors will find text in the reviews unpacking this comment

These are important points for the clarity of the manuscript, and we appreciate the Editor and Reviewers bringing them up. We apologize that the descriptions were not clear in the original manuscript. The key terminological distinction, which has not been clearly established in the literature, is between spindle rates reported at the single-electrode level (as in Gais et al., *J Neurosci*, 2002) and at the level of multiple electrodes, where spindles occurring simultaneously across multiple electrodes can be recognized as the same event. To clarify this distinction, we now define a “single-electrode spindle rate” (Figure 1—figure supplement 3) and compare this both to previous work (Gais et al. 2002, Fogel, S. M., et al. 2007 and Purcell, S. M., et al. 2017) and the “multi-electrode rate” (Figure 3b) computed over the entire array of electrodes. Because the multi-electrode spindle rate considers spindles occurring simultaneously on multiple electrodes as a single event and divides these events into one of three groups (local, regional, or multi-area), this measure will generally be lower than the single-electrode rate. We have now added a specific discussion in the manuscript on this point, along with further details, to clearly distinguish between these two measures and to highlight the results of our comparison (lines 225-230).

Furthermore, to provide additional insight on the increase in multi-area spindles following high-load visual memory tasks, we compare multi-electrode spindle rates between high- and low- load visual memory conditions. Comparing multi-electrode spindle rates demonstrates that, though multi-area spindles are indeed more rare events involving coordination across multiple brain areas, these events uniquely display a rather substantial and significant modulation across pre-sleep memory engagement (lines 231-234). We also note that our analyses earlier in this work indicate that the EEG datasets may contain fewer multi-area spindles (due to their decreased signal-to-noise ratio) than the iEEG (lines 612-615). Future work involving pre-sleep memory tasks during iEEG may have an opportunity to resolve even larger changes in these sleep events recorded at very high spatial and temporal resolution (lines 329-331).

Lastly, we appreciate the suggestion to use a non-parametric test by the Editor and Reviewer to assess statistical significance. While we initially took care in applying the t-test to these data, for example by verifying normality of the difference between paired observations with a Kolmogorov-Smirnov test before implementing the paired-sample t-test, we agree that a non-parametric test may be more suitable in this case. Following the Editor and Reviewer's suggestion, we switched to the paired-sample Wilcoxon signed-rank test to evaluate the statistical significance between low- and high-load conditions, with results unchanged from before (Figure 3b, Figure 3—figure supplement 1 and lines 231-232). To maintain consistency throughout the manuscript, we have now updated the statistical tests for analyzing rotating waves (Figure 4) to the paired-sample Wilcoxon signed-rank, with similar results as previously (lines 260-261, 263-264 and 266-267).

3. Tone down claims of generality of the CNN, and provide clear explanations for what makes the choice of architecture suitable to the current application to spindle detection. The reviewing editor finds this the most important piece of feedback from the reviews. As written, the reviewing editor feels that the manuscript may promote a culture of using deep learning in neuroscience without understanding how deep learning works. The authors have a unique opportunity of not only presenting an application of deep learning that leads to new scientific findings/hypotheses but also of doing so in a manner that promotes a culture of looking inside the black box and trying to understand it. The authors will find text in the reviews unpacking this comment

We appreciate the Reviewing Editor raising this important general point. Our goal in introducing the CNN technique is to provide a flexible and robust method to identify sleep spindles in noisy electrophysiological recordings, a problem where solutions have often relied on data manually scored by neurologists (expensive and subjective) or on amplitude-based automated techniques that miss many spindle events (as demonstrated here). Our goal with this technique, however, is by no means to encourage blind application of deep learning to this problem; rather, for each main result in the analysis we have utilized our two-step approach to create a validated and well-controlled process. To be specific, throughout the manuscript we (1) use the CNN method to detect a comprehensive set of spindles in the recordings and (2) validate the result obtained on the CNN-detected spindles on the subset of events detected by the more-conservative SNR method. In this way, in addition to providing a useful method to train the CNN on high-quality events that do not require manual detection (lines 133-135), we believe this two-step process represents a methodological advance that can make the most of the benefits of the deep learning approach (flexible waveform detection in difficult, high-noise settings) while also providing systematic validation of results obtained by the CNN on a high-quality subset (lines 86-96 and 103-105). In this way, we believe this work points towards a more general idea for application of deep learning techniques for neural data, allowing us to answer an important question in the sleep field: how do we make population-level statements about a set of neural events detected under uncertainty, without relying on arbitrary thresholds?

With this in mind, we certainly agree with the Reviewing Editor that application without understanding is a concern when introducing deep learning methods for neural data. This concern motivated our considerations throughout this work, and it is a concern that should remain central for applications of deep learning approaches in neuroscience. To this end, we have added a section in the manuscript specifically addressing this concern and highlighting the ways in which our two-step approach provides a flexible and rigorous solution to detecting specific neural events in electrophysiological recordings (lines 349-357). We have also added additional analyses to understand the choice of CNN architecture and hyperparameters, and features most important for spindle detection by the CNN model, including the filters and saliency maps for the trained CNN (new Figure 1—figure supplement 7 and 8), filter size of convolutional layers (new Figure 1—figure supplement 9), along with new supplementary sections “CNN choice of architecture and hyperparameter setting” and “CNN Visualization and interpretation” and additional text to discuss this point (lines 170-173). Finally, we have also added a section in the text carefully noting potential caveats in applications of deep learning for analysis of neural recordings and emphasizing the importance of well-controlled and interpretable analyses in this and future work (lines 357-372).

We believe that highlighting the necessity for well-controlled analyses and interpretability of these algorithms can address the concern on the motivation for deep learning in neuroscience. We believe it is possible to integrate deep learning methods into tightly controlled analysis workflows for electrophysiological and neuroimaging data, but agree that without proper controls, potential caveats of these methods would remain unaddressed. We hope the additional text and analyses fully address the concern of the Reviewing Editor here.

Reviewer #1:The manuscript has improved substantially and many of my concerns have been addressed. The deep-learning spindle detection method is more fully reported and the new controls with simulated data (varying noise and amplitude) are an improved approach to compare the CNN and AT methods (Figure S1, new Supplementary Table 1). The new analyses investigating the rotating spindle waves provide a framework for understanding the differences with prior reports of low synchrony.

We thank the Reviewer for the thoughtful comments and concerns and appreciate acknowledging our work to address them in the first round of revision. We take your new comments to heart and have worked to address them all in this new revision.

However, there are several important concerns that should be addressed.The new comparisons between CNN and AT models using surrogate data with systematically varied noise are useful. However, what was the rationale to compare the specific types of noise presented in Supplementary Table 1? Both the CNN and the AT have similar sensitivities and specificities for all noises, which may suggest that the different noise conditions do not constitute substantially different challenges for the detectors (also suggested by the example in Supplementary Figure 1, in which the spindle has a high signal to noise ratio with all types of noise). I think the readers will be more convinced of the value of the CNN method it can offer an advantage under a 'noise' condition in which the traditional AT methods are likely to struggle. For example, a more interesting source of noise may be the presence of non-spindle oscillations due to brief state changes (e.g., REM or α), or noise artifacts (e.g., examples in Supplementary Figure 12). The sensitivity of traditional methods is likely to go down in examples like these, how does the CNN compare?

We thank the Reviewer for raising this important point and suggesting a new analysis to provide additional evidence on the advantages of using the CNN model than the traditional AT approach. The logic behind this analysis was indeed to demonstrate improved performance of the CNN model under different noise types and magnitudes. Following this suggestion, we studied the change in the performance of the CNN and AT approaches under biological forms of noise, such as theta oscillations, and non-biological artifacts, such as line noise, electrical noise, and movement artifacts. We simulated theta oscillation using the approach proposed in the previous revision. We used Equation 1 in the paper with the oscillation angular frequency randomly selected from the 4-8 Hz frequency range to simulate theta (lines 652-656). To simulate non-biological artifacts, we first randomly chose a subset of artifacts detected as spindles by the AT approach in the iEEG recording. We then used the Fast Fourier Transform (FFT) to convert these artifacts into frequency domain. We next used the signal amplitude and randomly selected phases from [0, 2π], to generate a new set of artifacts and used the Inverse FFT to convert the signal back to time domain (lines 656-662). We visually inspected the simulated artifacts and verified the signals by comparing the PSD of the simulated artifacts with the original artifacts (new Figure 1—figure supplement 6a). We finally ran the CNN model and AT over both simulated datasets. Similar to previous cases, CNN model tends to have a higher performance than AT approach. However, the performance of AT drops significantly once we added noise artifact (Updated File S1).

We then studied the effect of artifacts on performance of the CNN and AT approach by systematically increasing the number of artifacts per minute in the surrogate data. In particular, the AT approach seems to be very sensitive to recording artifacts. The CNN model performs robustly in face of increases in the number of artifacts, while the performance of the AT gradually decreases as the number of artifacts per minute increases (new Figure 1—figure supplement 6b). This result further verifies that our approach is not sensitive to different types of artifact in the recording, as opposed to the AT approach (lines 672-678).

Taking these results together, we believe that, in contrast to the AT approach, which tends to be sensitive to recording artifacts, the CNN model performs robustly across a diverse range of artifacts and noise. These new analyses further confirm that the CNN approach introduced here can accurately detect spindles across a range of sleep recording types and provide better performance than the commonly used AT technique. We have added new figures (Figure 1—figure supplement 6) and analysis in supplementary section “Simulated data control varying noise” and lines 166-170 in the revised manuscript to further demonstrate that our deep learning approach is robust to different types of noise on the recording electrode.

Line 92: "co-occurrence"; please clarify the time window or time overlap used to determine co-occurrence. Did any amount of time overlap between spindles count as co-occurrence? Likewise, in Line 150: "simultaneously detected spindles", line 151: "based on co-occurrence". I think it's worth emphasizing in this paragraph that 'co-occurrence' and 'simultaneous' detection do not imply 0-lag synchrony (as discussed in the 'rotating waves' sections).

We thank the Reviewer for raising this point, and we appreciate this opportunity to make this point clear. Indeed, the two concepts of co-occurrence and zero-lag synchrony are distinct features of spindle events that are often not clearly distinguished in the literature. Here, by “co-occurrence” we indicate spindles that are detected simultaneously in different electrodes in the same 500-ms detection window. As mentioned by the Reviewer, this does not necessarily indicate the spindles also exhibit zero-lag synchrony, and we indeed find in our phase analyses that spindles systematically exhibit phase offsets across electrodes. In the revised submission, we have included a few sentences clarifying this point in the main text (lines 107, 119-121, 123-125 and 176).

Lines 102 and 104: "low and high visual memory task" should be "low and high-load…" (also in the Figure 3 legend, and in lines 261-262, 263: 'high-load' visual memory tasks).

Thank you for pointing this out. We have updated the revised manuscript accordingly.

There are some inconsistencies in the reported rates (spindles/min) across analyses and figures: Suppl. Figure 4a indicates average spindles/min around 4-5/min in all data types, including EEG data used in the visual memory task. However, the average from the distributions shown in Figure 3b seems much lower, even when combining all spindles across each memory condition, e.g. for the H-VM is about 1/min in local and regional and about 0.5/min for multi-area. In other words, could you clarify how the averages of the distributions in Figure 3b add up to 4 spindles/min in Figure 4a?

These are important points for the clarity of the manuscript, and we appreciate the Reviewer bringing them up. We apologize that the descriptions were not clear in the original manuscript. As noted for Main Critique 2, the key distinction here is between spindle rates reported at the single-electrode level (as in Gais et al., *J Neurosci*, 2002) and at the level of multiple electrodes, where spindles occurring simultaneously across multiple electrodes can be recognized as the same event. To clarify this distinction, we now define an “single-electrode spindle rate” (Figure 1—figure supplement 3) and compare this both to previous work (Gais et al. 2002, Fogel, S. M., et al. 2007 and Purcell, S. M., et al. 2017) and the multi-electrode spindle rate (Figure 3b) computed over the entire array of electrodes. Because the multi-electrode spindle rate considers spindles occurring simultaneously on multiple electrodes as a single event and divides these events into one of three groups (local, regional, or multi-area), this measure will generally be lower than the single-electrode rate. We have now added a specific discussion in the manuscript on this point, along with further details, to clearly distinguish between these two measures and to highlight the results of our comparison (lines 225-230).

In addition, in Figure 3b, within the low-VM condition the rate of multi-area spindles is substantially lower (< 0.5/min) than local and regional (about 1/min on average); and within high-VM the rate also appears much lower (<1/min) for multi-area compared to the distributions for local and regional spindles (up to ~2/min). The text indicates p-values for what seems to be a comparison of relative rates (between H-VM and L-VM), which is interesting but is this result due to a significantly lower multi-area spindle rate with low memory? Based on the figure it looks like multi-area spindles occur at lower rates regardless of memory load? It will be helpful to provide statistical comparisons of spindle rates (among all spindle types) within each condition (L-VM, H-VM) so that the relative changes with memory can be interpreted in the context of the absolute spindle rates in each memory condition. Based on the distributions it seems that non-parametric tests would be more appropriate than t-tests.

We thank the Reviewer for bringing up these important points, which we believe are an opportunity to provide additional insight on the increase in multi-area spindles following high-load visual memory tasks. Importantly, we note that our comparison was always between the absolute rate of spindles in the L-VM and H-VM conditions. For example, the detection rate of multi-area spindles in the L-VM condition is 0.38 ± 0.02 detections/minute (average across subjects ± SEM). In H-VM, however, this rate increases to 0.62 ± 0.08 detections/minute, almost a doubling of the rate. By contrast, the rate of local spindles in the L-VM condition is 0.77 ± 0.10 detections/minute, with a similar rate of 0.79 ± 0.11 detections/minute in H-VM. This comparison of absolute multi-electrode spindle rates demonstrates that, though multi-area spindles are indeed more rare events, involving coordination across multiple brain areas, these events uniquely display a significant and rather substantial modulation across pre-sleep memory engagement. We appreciate the opportunity to clarify this point, as the comparison here was not sufficiently clear in the previous version. We do note that the rates estimated in this work include partially overlapping windows; however, because our comparison in this work is focused on the difference between L-VM and H-VM, this does not bear relevance to the differences reported here (as verified in additional subsequent analyses with varying window offset). Finally, we also note that our analyses earlier in this work indicate that the EEG datasets may contain fewer multi-area spindles (due to their decreased signal-to-noise ratio) than the iEEG (lines 612-615). Future work involving pre-sleep memory tasks during iEEG may have an opportunity to resolve even larger changes in these sleep events recorded at very high spatial and temporal resolution (lines 329-331). We again thank the reviewer for pointing out the importance of stating the absolute spindle rates, which we have now included in lines 232-234 and 544-549 of the text.

We also appreciate the suggestion to use a non-parametric test by the Reviewer to assess statistical significance. While we initially took care in applying the t-test to these data, for example by verifying normality of the difference between paired observations with a Kolmogorov-Smirnov test before implementing the paired-sample t-test, we agree that a non-parametric test may be more suitable in this case. Following the Reviewer's suggestion, we switched to the paired-sample Wilcoxon signed-rank test to evaluate the statistical significance between low- and high-load conditions, with results unchanged from before (Figure 3b, Figure 3—figure supplement 1 and lines 231-232). To maintain consistency throughout the manuscript, we have now updated the statistical tests for analyzing rotating waves (Figure 4) to the paired-sample Wilcoxon signed-rank, with similar results as previously (lines 260-261, 263-264 and 266-267).

Suppl. Figure 4: missing (a),(b) labels.

Thanks for catching the missing labels. We have added the labels in the revised manuscript.

Supplementary Figure 4a is not described in the text?

We thank the Reviewer for noting this point and apologize that this figure (Supplementary Figure 4a, now updated to Figure 1—figure supplement 3a) was not described in the manuscript. We have added a reference and detailed description for Figure 1—figure supplement 3a in the main text (lines 156-157 and 345-346)

Line 725: 12? sites

Thanks for catching this typo. We have corrected the error in the revised manuscript (Figure 1—figure supplement 4**)**.

Line 154: it'd be useful to reference the Supplementary Table 2 (with the list of the cortical regions).

We appreciate this suggestion, and we have added a reference to File S2 (previously called Table S2) in the revised manuscript (line 181).

Lines 170, 171: I believe the verb should be in past tense.

We apologize for the inconsistent verb tenses. We have corrected the verb tenses throughout the revised manuscript.

Line 179: The interpretations regarding the association between the visual memory load and subsequent sleep spindles is limited to memory consolidation processes. However, memory performance after sleep was not studied in this work, therefore "the impact on memory consolidation" cannot directly be assessed. Sentences such as these (lines 179-180): "If this is the case, what is the impact of the distribution on sleep-dependent memory consolidation? To answer this question…" could be rephrased to state the key question that the data can address. Other sentences in the manuscript are more accurate and valuable in this respect, e.g., Lines 289-290 "(2) this spatial extent can be modulated by the specific memory conditions prior to sleep (Figure 3)". Indeed, memory consolidation is one of several mechanisms that determine how wakefulness influences sleep oscillations. The authors correctly cover literature on the association between memory and spindles "An increase in spindle density after memory tasks and its relationship with memory consolidation is well established (Clemens et al., 2005; Dang-Vu et al., 2008; Gais et al., 2002; Schabus et al., 2007, 2004)", but the impact of the authors' findings would be enhanced by discussing other hypotheses in the sleep field that are also consistent with the presented results; for example, the high-load working memory condition may increase firing rates and entrain downscaling processes during subsequent sleep (Tononi and Cirelli, 2014; Crunelli et al., 2018; Klinzing et al.,2019).

We thank the Reviewer for bringing up this important point. In our revised manuscript, we have updated the language to more carefully focus on the impact of pre-sleep memory tasks on the distribution of spindles in cortex (lines 72, 209 and 273-275). We also appreciate the Reviewer’s point that these results could be consistent with the process of synaptic downscaling. We have included a more detailed treatment of this point in the Discussion (lines 289-295), along with specific suggestions for how future work could address these hypotheses by studying sleep recordings (lines 317-318). We again thank the Reviewer for this suggestion and hope the additions to the manuscript fully address this point.

Paragraph starting 215: is this only with the EEG data?

Yes, this analysis is only performed on the EEG data which had the unique feature of testing sleep after tasks with varying memory loads. In this analysis, we are focusing on the comparison of spatiotemporal dynamics in low-load and high-load memory tasks. We have now noted this more clearly in the revised text (lines 121 and 253).

Figure 4a: missing (a),(b) labels for the plots. Indicate the x,y value of the removed outlier point in parenthesis (in 'a' and 'c') since it's not in the plot but still used in quantifications.

Thanks for catching the missing labels and the suggestion. We have added the outliers and labels in the revised manuscript (Figure 4a, b and c).

Line 552: "We simulated 60-minutes recording containing" should it be "60-minutes of recording…"?

Thanks for catching this typo. We have corrected it in the revised manuscript (line 617).

Reviewer #2:The authors suggest that the lack of algorithms for reliably identifying spindles in neural recordings has led to the under reporting/underestimation of the spatial extent to which spindles occur in the brain. The authors propose an approach based on CNNs that they claim can detect spindles more reliably than existing ones, lead to new insights as to how cross-region spindles may contribute to the integration of 'information' across brain areas.Strengths1. The applicability of the proposed methods to neural data from multiple modalities, i.e. EEG, EcOG and iEEG.2. The framework the authors lay out for constructing high-quality data sets to train the CNNs.3. The combination of computational methods and new suggested insights into how spindles can facilitate the integration of 'information' across brain areas.

We thank the Reviewer for highlighting the novelty and strength of our submission and appreciate the opportunity to discuss these points.

Weaknesses1. The authors could improve the explanations as to why the CNN seems to do better than legacy methods such as the AT. The explanation surrounding the CNN's ability to detect spindles of different amplitudes does not seem satisfactory enough.

We appreciate the opportunity to further discuss these important points. One of the main advantages of the CNN model is its ability to detect clearly-formed spindles ranging from lower to high amplitudes. Thus, while AT approaches tend to detect only the highest-amplitude spindle events (as in Figure 1—figure supplement 4), the CNN detects a broader set of spindles. The second advantage of the CNN model is its ability to generalize well across recordings with different types of noise, provided there is sufficient training data and high-quality marked events. The results from our previous analysis indicate that our CNN approach can achieve a very high performance when trained under varying noise and recording conditions (Figure 1—figure supplement 5 and File S2). In this revision, we also study change in the performance of the CNN and AT approaches under biological forms of noise such as the REM theta oscillation and non-biological artifacts. Under these conditions, the CNN approach clearly outperforms the AT, due in large part to the linear decrease of AT performance with increase in the frequency of artifacts (lines 672-678, and Figure 1—figure supplement 6b). Taking these results together, we believe the evidence in this work demonstrates the CNN model detects a broader range of spindles and performs robustly across a diverse range of artifacts and noise as opposed to the AT approach. We have added additional explanation in lines 166-170 and supplementary section “Simulated data control – varying noise” in the revised manuscript to fully address the Referee’s concern on this point.

2. The authors claim the applicability of the CNN to data from different modalities, and its generality, as a strength. Given the black-box nature of CNNs, and the lack of an attempt in the manuscript to explain the CNN, its success/failure modes, which aspects of the data it focuses on to detect spindles (e.g. saliency maps), these claims do not feel justified to the reviewer and may contribute to the proliferation of black-box CNNs in neuroscience

We thank the Referee for bringing up this important point, which we have considered very seriously in the process of revision. Our goal in introducing the CNN technique is to provide a flexible and robust method to identify sleep spindles in noisy electrophysiological recordings, a problem where solutions have often relied on data manually scored by neurologists (expensive and subjective) or on amplitude-based automated techniques that miss many spindle events (as demonstrated here). Our goal with this technique, however, is by no means to encourage blind application of deep learning to this problem; rather, for each main result in the analysis we have utilized our two-step approach to create a validated and well-controlled process.

With this in mind, we certainly agree with the Reviewer that application without understanding is a concern when introducing deep learning methods for neural data. This concern motivated our considerations throughout this work, and it is a concern that should remain central for applications of deep learning approaches in neuroscience. To this end, we have added a section in the manuscript specifically addressing this concern and highlighting the ways in which our two-step approach provides a flexible and rigorous solution to detecting specific neural events in electrophysiological recordings (lines 349-372). We have also added additional analyses to understand the choice of CNN architecture and hyperparameters, and features most important for spindle detection by the CNN model, including the filters and saliency maps for the trained CNN (new Figure 1—figure supplement 7 and 8), filter size of convolutional layers (new Figure 1—figure supplement 9), along with new supplementary sections “CNN choice of architecture and hyperparameter setting” and “CNN Visualization and interpretation” and additional text to discuss this point (lines 170-173).

We believe that highlighting the necessity for well-controlled analyses and interpretability of these algorithms can address the Reviewer’s concern on the motivation for deep learning in neuroscience. We believe it is possible to integrate deep learning methods into tightly controlled analysis workflows for electrophysiological and neuroimaging data, but agree that without proper controls, potential caveats of these methods would remain unaddressed. We hope the additional text and analyses fully address the concern of the Referee here.

Recommendations for the authors:The reviewing editor found the manuscript a much improved version of the initial submission.1. The reviewing editor found that the analyses currently in the manuscript do not support the authors' claims of generality of the CNN (e.g. lines 376-377), and that the language surrounding these claims may contribute to supporting an already-widespread tendency to utilize black-box neural networks for analyzing neural data.a) One feedback from the first round of reviews had to do with the lack of details surrounding the CNN. The reviewing editor appreciates the authors' attempt to improve this. The authors seem to want to emphasize the generality of the architecture. Such claims of generality do not give insight to the reader on how to pick an architecture for detecting oscillatory patterns in a different context (e.g. ripples). The success of the architecture for gravitational wave detection, its use in the current manuscript, do not give license to use it w/o changes in any application.A practitioner would appreciate guidance on architecture design. Given the length of pattern of interest and a sampling rate, a user would want to know how to pick filter sizes. For instance, for a patter of length 0.5 seconds, a 256 Hz sampling rate, the effective filter (explained below) associated with an architecture ought to have size on order of 100 samples (0.5*256 = 128 samples), not 10 or 1000. The reviewing editor thinks this manuscript could have a much stronger impact if the authors took such questions into considerations. Have authors visualized the filters they learn? Have they considered generating saliency maps to determine which parts of inputs the network focuses on to detect spindles? At present, the emphasis on the generality of the CNN, w/o considerations for what about the authors' specific context makes the choice of architecture suitable (other than the fact that it works for gravitational-way detection) feels a bit disappointing.More details on guidance on how to design architecture/why current architecture works: For any conv net, the choice of filter size at each layers, together with number of layers relates to size of features one would like to detect. Each layer a conv. Composition of conv equals a conv with a filter size roughly proportional to the sum of filter sizes at each layers. 7x7 filters not uncommon in image proc. Cascading 3-4 such layers (ignoring nonlinearities) gives a filter with an effective 30x30 size.Concrete suggestions: (a) adding text to the manuscript explaining what what about the authors' specific context makes the choice of architecture suitable (other than the fact that it works for gravitational-way detection),

We thank the Reviewer for noting that more details are necessary to clarify the choice of CNN architecture and we apologize that our explanation was not clear in the previous revision. The CNN architecture should be tailored with respect to the duration of rhythmic activity, type of oscillation and sampling rates of recording modality. The duration of the rhythmic activity and sampling rate determine the length of the sliding window for the CNN model. For example, in our NHP EEG recording with the sampling rate of 1000 Hz, the sliding window is set to 0.5 sec which is the minimum duration of spindle activities observed during sleep and contains 500 data points. We then specify the filter size with respect to the length of the sliding window and types of rhythmic activity. The CNN layer is designed to start by extracting local features, gradually extracting longer-timescale features by decreasing the feature space. Filter sizes covering up to approximately one oscillation cycle (70-120 msec) are effective in detecting spindle activity. To understand this further, we simulate 10 recordings with 2 spindles per minute and add different types of noise including the theta oscillation and artifact. Using these surrogate recordings for which we have the exact timing of spindles, we demonstrate that longer filter size is ineffective at detecting spindles. Specifically, we gradually increase the filter size (new Figure 1—figure supplement 9) and compute the performance of the CNN model. As expected, the CNN performance drops as we increase the filter size, in accordance with this underlying mechanism. This result further validates the generality of the CNN approach for detecting neural rhythms, while also getting at the mechanism. We believe that a similar mechanism can be implemented for specifying the filter size for other neural and biological rhythms. The current CNN architecture works well with slight changes in the sliding window (duration and sampling rate) and type of oscillation, but it requires modification otherwise.

We have added a new supplementary section “CNN choice of architecture and hyperparameter setting” and figure (Figure 1—figure supplement 9) to explain the underlying mechanism of setting filter size in the CNN model. We hope these additional clarifications fully address the Referees’ concerns on this point. We are also providing a guideline on how to set the parameters of the CNN model in our toolbox for different detection settings. The first GitHub version of the toolbox is attached for reference.

b) visualizing filters learned by the architecture, c) consider generating saliency maps.

We thank the Reviewer for these suggestions. To understand this point further, we studied the filter and saliency maps of the CNN models. To do this, we first simulate six spindle/non-spindle signals and visualize the feature and gradient maps corresponding to these input signals. The feature map can accurately detect the maximum amplitude within each cycle using the maximum activation of the CNN model across all oscillations (new Figure 1—figure supplement 7). Using these features, which specify the timing and relative height of the detected maximum amplitudes, the CNN model can learn to reliably detect the specific waveform characteristics of the sleep spindle in different types of recordings. In addition, we studied gradient attribution maps to identify part of the signal that are most important for classification in the CNN model (new Figure 1—figure supplement 8). The area of the signal with the highest modulation in amplitude has the greatest impact on the classification. The pattern of activations and gradient map across the simulated signals provides insights into the underlying mechanism by which the CNN model efficiently distinguishes between different types of oscillation. We thank the Reviewer for pointing out the importance of including more details on how the CNN model works. We have added a few sentences in the main text (lines 170-173), a new supplementary section, “CNN Visualization and interpretation” and figures (Figure 1—figure supplement 7 and 8) in the revised manuscript.

b) The reviewing editor feels that the authors can improve the section detailing the training of the CNN. The editor suggests the authors consider using a table to summarize different aspects (number of layers, filter sizes etc….). The editor also suggests that, very early on, the authors mention the subject-dependence of the training.

We appreciate the Reviewer’s comment on the training aspect of the CNN model. As requested in the previous round of revision, we made a grid search over the potential range of hyperparameters and measured the detection similarity using Cronbach's α. New File S3 contains the details of our sensitivity analysis in the updated manuscript. We also performed further analysis on the choice of CNN architecture and filter size in the new supplementary section, “CNN choice of architecture and hyperparameter setting”. The result from our sensitivity analysis confirms that the proposed CNN architecture has high reproducibility under moderate parameter variability.

Finally, following the Reviewer’s suggestion, we added a few sentences early on in the main manuscript (lines 100-104 and 132-133) explaining that the CNN model is subject-dependent. We appreciate your comments to improve the paper and we have added additional clarification in the revised manuscripts to answer your concerns.

2. Misc questionsa) In supplementary figure 4b, the scatter plot for EEG (green) suggests a similar max PSD for the CNN and AT. I would expect a similar rate of spindle detection. Supplementary figure 4a suggests otherwise. Can the authors explain?

We thank the Reviewer for noting this point. While the AT approach consistently detects spindles of higher amplitude than the CNN in the iEEG and ECoG recordings, this effect is indeed not observed in the EEG recordings. We believe this may be due to lower overall signal-to-noise ratio in the EEG; however, it is difficult to determine the precise cause underlying this change in these two datasets. We have noted this point in the text and suggested possible analyses in simultaneous EEG-iEEG recordings (which are often taken by our clinical colleagues at LHSC) for future work (lines 317-318 and 612-615).

b) In Figure 2c, EcOG the fact that the CNN detects fewer local and regional spindles per minute than the AT method requires explanation. Does this figure refer to correctly detected spindles? Should the editor interpret the lower ratio as the AT method having more false alarms than the CNN?

We thank the Reviewer for noting this point. For one subject in the ECoG recordings analyzed here, the distribution of signal amplitudes in the 11-15 Hz frequency band varies so widely that a fixed threshold in the AT technique no longer distinguishes well between spindle and non-spindle activity at all electrodes. The variation in amplitude across electrodes results in incorrect detections in electrodes with lower amplitudes and missed detections in electrodes with higher amplitudes. For this reason, the AT technique in this subject caused more detections of local and regional spindles. Importantly, this point further demonstrates the potential problems with the AT approach, where artifacts or recording noise can significantly affect results on spindle distribution or occurrence. We appreciate the Reviewer’s comment and have now updated the manuscript to explain this point (lines 196-199).